RESEARCH COMMUNICATION

# IgE-mediated mast cell activation promotes inflammation and cartilage destruction in osteoarthritis

Qian Wang[1,2†], Christin M Lepus[1,2†], Harini Raghu[1,2†], Laurent L Reber[3‡], Mindy M Tsai[3], Heidi H Wong[1,2], Ericka von Kaeppler[1,2], Nithya Lingampalli[1,2], Michelle S Bloom[1,2], Nick Hu[1,2], Eileen E Elliott[1,2], Francesca Oliviero[4], Leonardo Punzi[4], Nicholas J Giori[1,5], Stuart B Goodman[5], Constance R Chu[1,5], Jeremy Sokolove[1,2], Yoshihiro Fukuoka[6], Lawrence B Schwartz[6], Stephen J Galli[3,7], William H Robinson[1,2]*

[1]GRECC, VA Palo Alto Health Care System, Palo Alto, United States; [2]Division of Immunology and Rheumatology, Stanford University School of Medicine, Stanford, United States; [3]Department of Pathology, Stanford University School of Medicine, Stanford, United States; [4]Rheumatology Unit, Department of Medicine, University of Padova, Padova, Italy; [5]Department of Orthopedic Surgery, Stanford University School of Medicine, Stanford, United States; [6]Department of Internal Medicine, Virginia Commonwealth University School of Medicine, Richmond, United States; [7]Department of Microbiology and Immunology, Stanford University School of Medicine, Stanford, United States

**\*For correspondence:**
w.robinson@stanford.edu

[†]These authors contributed equally to this work

**Present address:** [‡]Center for Physiopathology of Toulouse-Purpan (CPTP), UMR 1043, University of Toulouse, INSERM, CNRS, Toulouse, France

**Abstract** Osteoarthritis is characterized by articular cartilage breakdown, and emerging evidence suggests that dysregulated innate immunity is likely involved. Here, we performed proteomic, transcriptomic, and electron microscopic analyses to demonstrate that mast cells are aberrantly activated in human and murine osteoarthritic joint tissues. Using genetic models of mast cell deficiency, we demonstrate that lack of mast cells attenuates osteoarthritis in mice. Using genetic and pharmacologic approaches, we show that the IgE/FcεRI/Syk signaling axis is critical for the development of osteoarthritis. We find that mast cell-derived tryptase induces inflammation, chondrocyte apoptosis, and cartilage breakdown. Our findings demonstrate a central role for IgE-dependent mast cell activation in the pathogenesis of osteoarthritis, suggesting that targeting mast cells could provide therapeutic benefit in human osteoarthritis.
**Editorial note:** This article has been through an editorial process in which the authors decide how to respond to the issues raised during peer review. The Reviewing Editor's assessment is that all the issues have been addressed (see decision letter).
DOI: https://doi.org/10.7554/eLife.39905.001

## Introduction

Osteoarthritis, characterized by progressive degeneration of articular cartilage in the joints, is a major cause of disability and the most common form of arthritis (*Felson, 2006*). Current treatment approaches are limited to pain reduction and joint replacement, highlighting the importance of understanding the mechanisms underlying the pathogenesis (*Chevalier et al., 2013*; *Wieland et al., 2005*). While low-grade synovial inflammation is a widely recognized feature of osteoarthritis (*Atukorala et al., 2016*; *Hill et al., 2007*; *Robinson and Mao, 2016*; *Sellam and Berenbaum, 2010*), the underlying cellular and molecular mechanisms are not fully defined. Emerging evidence

suggests that dysregulated activation of innate immunity involving macrophages and mast cells are likely involved in the pathogenesis of this disorder (*de Lange-Brokaar et al., 2016*; *Kraus et al., 2016*; *Liu-Bryan and Terkeltaub, 2015*; *Raghu et al., 2017*).

Mast cells are sentinels of the innate immune system, poised to rapidly respond to exogenous pathogens and to endogenous danger signals (*Bischoff, 2007*). A wide variety of stimuli (e.g., allergens that cross-link IgE-bound high affinity IgE receptor (FcεRI) or antibodies that directly cross-link FcεRI, cytokines such as IL-33, complement anaphylatoxins, immune complexes, neuropeptides, TLR ligands, etc.) can influence mast cell degranulation and release of pre-formed mediators including histamine, tryptases, pro-inflammatory lipids, cytokines and chemokines (*Theoharides et al., 2012*; *Yu et al., 2016*). Importantly, different activation stimuli are capable of inducing distinct mast cell responses in both physiological and pathological settings (*Enoksson et al., 2011*; *Gaudenzio et al., 2016*). In allergic disease—a setting in which mast cells have been most extensively studied—these mediators promote chronic allergic inflammation which, if sustained, results in long-term tissue damage, fibrosis, and remodeling (*Galli and Tsai, 2012*). Similar to the tissue remodeling in allergic diseases, human osteoarthritis and experimental osteoarthritis in rodents are characterized by abnormal and progressive bone and other tissue remodeling (*Remst et al., 2014*).

Several studies have documented the presence of mast cells and their mediators in the synovium and synovial fluids of individuals with osteoarthritis (*Buckley et al., 1997*; *Dean et al., 1993*; *Lee et al., 2013*). Recently, it was reported that synovial mast cell numbers and degranulation status correlate positively with increased synovitis and cartilage damage in patients with knee osteoarthritis, suggesting that mast cells might contribute to the pathogenesis of osteoarthritis (*de Lange-Brokaar et al., 2016*). Nevertheless, the precise role of mast cells in the pathogenesis of osteoarthritis has not been defined. Here, we provide evidence that demonstrates a pathogenic role for IgE-dependent mast cell activation and the mast cell mediator tryptase in osteoarthritis.

## Results

### Enhanced mast cell tryptase release, degranulation, and activation in osteoarthritis

Guided by knowledge that mast cells are present in osteoarthritic synovium (*Buckley et al., 1998*; *de Lange-Brokaar et al., 2016*; *Lee et al., 2013*), we analyzed synovial fluids for the mast cell-specific product, tryptase. We compared tryptase levels in the synovial fluids from individuals with osteoarthritis with those from non-osteoarthritis controls with prior joint trauma >6 months prior to sample collection but no radiographic osteoarthritis. Using Tosyl-Gly-Pro-Lys-pNA-based quantification, we found significantly elevated levels of catalytically active tryptase in synovial fluids from individuals with osteoarthritis as compared to non-osteoarthritis controls (*Figure 1a*). We also directly visualized mast cell degranulation in osteoarthritis by performing immuno-electron microscopy on synovial tissue sections stained with gold particle-labeled anti-tryptase antibody. Mast cells exhibiting features including tryptase-containing granule matrices located outside of the plasma membrane and/or fusion of granule and plasma membranes were identified as actively degranulating or degranulated (*Figure 1b*). We found significantly increased percentages of degranulated mast cells in osteoarthritic synovial linings compared to those from non-osteoarthritic joints (*Figure 1c*). Immuno-electron microscopy of these sections with a gold-labeled isotype-matched control antibody did not result in positive staining of mast cells (*Figure 1—figure supplement 1*), confirming the specificity of the anti-tryptase staining of mast cells. Nevertheless, anti-tryptase staining of osteoarthritic and non-osteoarthritic synovial linings revealed no significant differences between the numbers of mast cells present in these samples (*Figure 1—figure supplement 2*). Together, these findings demonstrate that mast cells are actively degranulating to release tryptase in osteoarthritic joints.

We also analyzed the gene expression of mast cell-related surface receptors, chemoattractants, and degranulation products in synovial membranes from individuals with early- or end-stage osteoarthritis and from healthy synovium. Unsupervised clustering of genes involved in mast cell survival, function or activation revealed two main clusters – osteoarthritis and healthy comparator synovium – with a statistically-significant broad upregulation of mast cell-related genes in the osteoarthritic relative to healthy synovium (*Figure 1—figure supplement 2a*). Supervised clustering of genes grouped based on their known function and segregated by disease stage of osteoarthritis revealed that genes

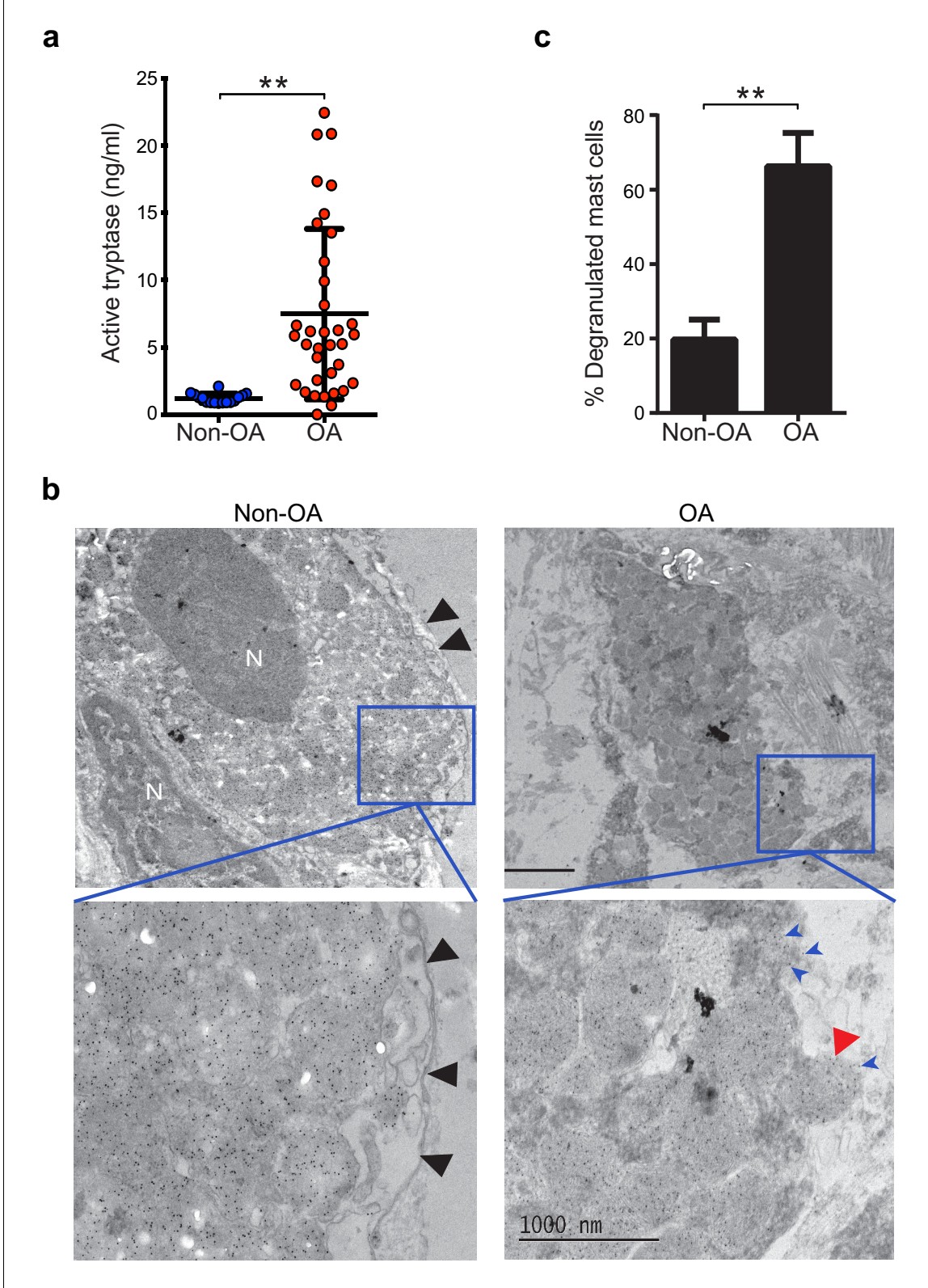

**Figure 1.** Increased mast cell degranulation and tryptase release in osteoarthritis. (a) Tosyl-Gly-Pro-Lys-pNA-based quantification of active tryptase in synovial fluids from individuals with osteoarthritis (OA; $n$ = 35) and from individuals with prior joint trauma but no radiographic osteoarthritis (PT non-OA; $n$ = 16). Bars represent mean ± s.d. **p≤0.01 by Mann-Whitney test, and results are representative of the results of three independent experiments performed using two independent sample sets. (b) Representative transmission electron microscopy images of osteoarthritic and non-osteoarthritic

*Figure 1 continued on next page*

*Figure 1 continued*

synovial tissue sections immuno-labeled with a gold-conjugated anti-tryptase antibody. Left panels: A quiescent mast cell with many cytoplasmic granules exhibiting strong immunoreactivity for tryptase and an intact plasma membrane (black arrowheads) in non-osteoarthritic synovial lining (Non-OA). Right panels: A degranulated mast cell exhibiting an exteriorized granule matrix with tryptase immunoreactivity (red arrowhead) in an osteoarthritic synovial lining (OA). There is also some other tryptase immunoreactivity apparent outside of this cell (blue arrowheads), likely derived from exteriorized granule matrices. Lower panels are higher magnification (8000×) images of area shown in the blue box in the corresponding upper panels (1500×). (c) Percentage of degranulated mast cells in synovial tissues obtained from individuals with osteoarthritis (n = 5) and non-osteoarthritis (n = 5). Intact and degranulated mast cells were counted by an examiner blinded to sample group assignment. Data are mean ± s.d. \*\*p<0.01 by Student's *t*-test, and are representative of three independent experiments using independent sample sets.

DOI: https://doi.org/10.7554/eLife.39905.002

The following figure supplements are available for figure 1:

**Figure supplement 1.** Transmission electron microscopy (TEM) isotype control immuno-labelling analysis.

DOI: https://doi.org/10.7554/eLife.39905.003

**Figure supplement 2.** Mast cells are present in human osteoarthritis and post-trauma non-osteoarthritis synovial tissue.

DOI: https://doi.org/10.7554/eLife.39905.004

**Figure supplement 3.** Expression of mast cell mediators in osteoarthritic synovial membranes.

DOI: https://doi.org/10.7554/eLife.39905.005

involved in mast cell proliferation and survival (e.g., *KIT* and *IL3RA*), protease processing and/or stabilization (e.g., *SRGN* and *CTSB*), and Fc receptor subunits (e.g., *FCER1A* and *FCER1G*) were significantly upregulated in the synovium of both early- and end-stage osteoarthritis compared to the healthy synovium (*Figure 1—figure supplement 2b*). Further, the expression of genes encoding pre-formed mediators such as proteases (e.g., tryptase-encoding genes *TPSAB1*, *TPSB2* and *TPSD1*) were likewise upregulated in osteoarthritic as compared to healthy synovial membranes (*Figure 1—figure supplement 2b*). These findings suggest that mast cells are transcriptionally active in osteoarthritic synovial tissues.

## Genetic elimination or pharmacologic inhibition of mast cells attenuates osteoarthritis

To evaluate whether mast cells directly participate in the pathogenesis of osteoarthritis, we surgically induced osteoarthritis through destabilization of the medial meniscus (DMM) (*Glasson et al., 2007*; *Loeser et al., 2013*) in mice lacking mast cells. We used two distinct mouse models of mast cell deficiency: 1) C57BL/6J-*Kit*$^{W-sh/W-sh}$ (*Kit*$^{W-sh/W-sh}$) mice (*Grimbaldeston et al., 2005*), which have a large gene inversion that results in reduced expression of c-kit, the receptor for the major mast cell growth factor stem cell factor, and 2) *Cpa3-Cre;Mcl-1*$^{fl/fl}$ (Hello *Kit*ty) mice, a c-kit-independent model of mast cell deficiency (*Reber et al., 2012*). Deficiency of mast cells in either model conferred significant protection against osteoarthritis-related pathologies (*Figure 2a–d*, *Figure 2—figure supplement 1*, and *Figure 2—figure supplement 2*). Twenty weeks after DMM surgery, cartilage loss, osteophyte formation, and synovitis were significantly reduced in *Kit*$^{W-sh/W-sh}$ mice compared to their age-matched, mast cell-sufficient littermate controls (C57BL/6J mice) (*Figure 2a–d*, *Figure 2—figure supplement 2*). We validated this observation in the *Cpa3-Cre;Mcl-1*$^{fl/fl}$ mice, which also developed less severe cartilage loss, osteophyte formation, and synovitis 20 weeks after DMM surgery (*Figure 2—figure supplement 1*).

Mast cell-deficient *Kit*$^{W-sh/W-sh}$ and *Cpa3-Cre;Mcl-1*$^{fl/fl}$ mice have phenotypic abnormalities in addition to their mast cell deficiencies. For example, *Kit*$^{W-sh/W-sh}$ mice have increased levels of circulating neutrophils and basophils, while *Cpa3-Cre;Mcl-1*$^{fl/fl}$ mice have reduced numbers of basophils (*Lilla et al., 2011*; *Reber et al., 2012*; *Tsai et al., 2005*). To ascertain whether the reduction in osteoarthritis-related pathology in *Kit*$^{W-sh/W-sh}$ and *Cpa3-Cre;Mcl-1*$^{fl/fl}$ mice was in fact due to the absence of mast cells, we engrafted bone marrow-derived mast cells into *Kit*$^{W-sh/W-sh}$ and *Cpa3-Cre;Mcl-1*$^{fl/fl}$ mice to generate mast cell-sufficient mice. Toluidine blue staining confirmed the presence of mast cells within synovium derived from C57BL/6J and *Cpa3-Cre;Mcl-1*$^{+/+}$control mice and within the synovium of mast cell-engrafted mice, whereas no mast cells were detected in most mast cell-deficient mice (*Figure 2—figure supplement 3a and c*). Quantification of toluidine blue-stained mast cells in the synovium derived from these mice demonstrated significant reductions in mast cell numbers in the synovium of *Kit*$^{W-sh/W-sh}$ and *Cpa3-Cre;Mcl-1*$^{fl/fl}$ mice as compared to the C57BL/6J

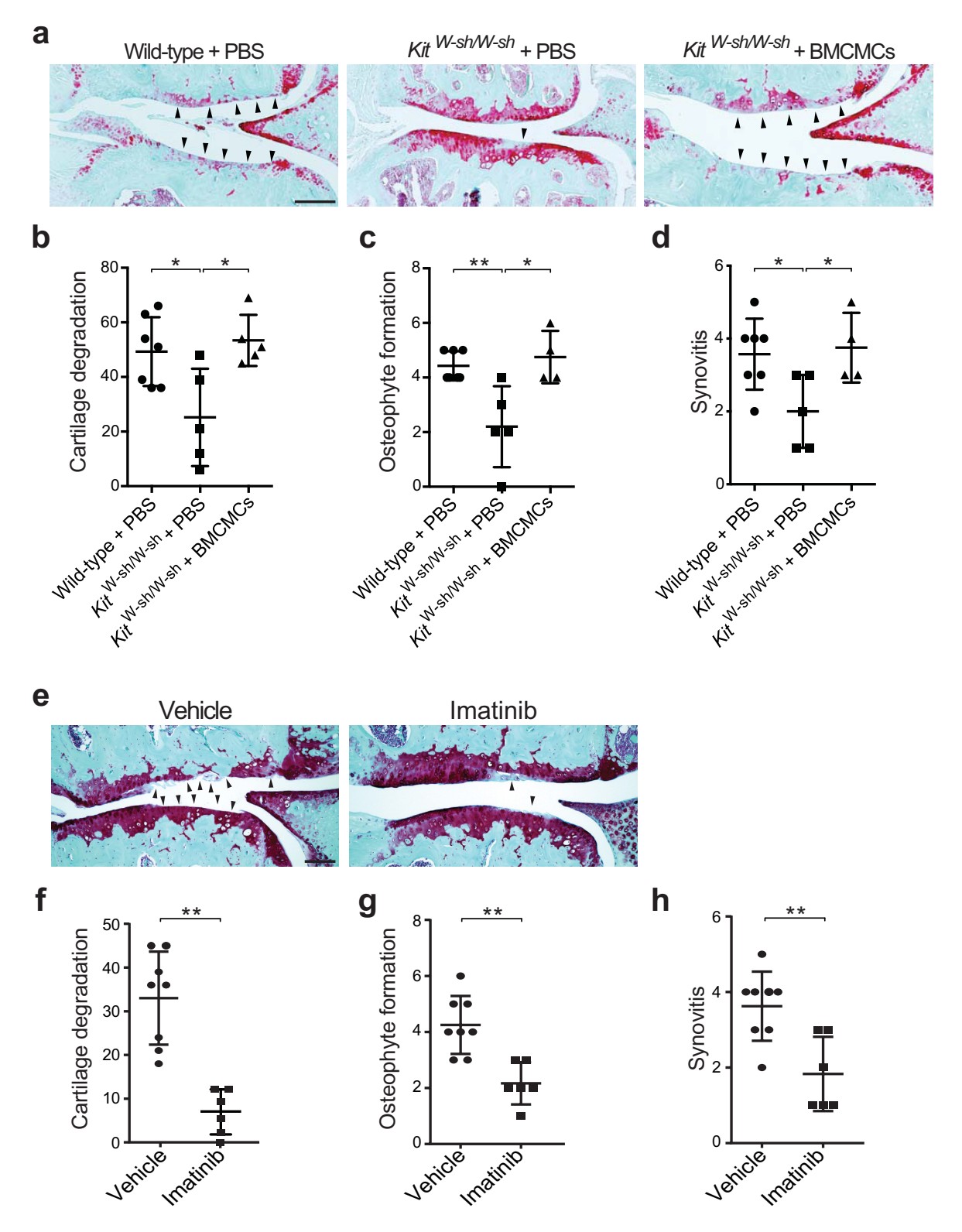

**Figure 2.** Genetic deficiency or pharmacologic inhibition of mast cells protects against the development of osteoarthritis in mice. (a–d) Cartilage degradation in medial regions of stifle joints from C57BL/6J mast cell-sufficient mice (Wild-type +PBS; $n = 7$), mast cell-deficient mice ($Kit^{W-sh/W-sh}$ + PBS; $n = 5$), and mast cell-deficient mice engrafted with BMCMCs ($Kit^{W-sh/W-sh}$ + BMCMCs; $n = 5$) 20 weeks after DMM surgery. Representative Safranin-O-stained sections of medial regions of stifle joints from these mice are shown (a); arrowheads show severe cartilage loss. Cartilage degradation (b),
*Figure 2 continued on next page*

*Figure 2 continued*

osteophyte formation (c), and synovitis (d) in medial regions of stifle joints from these mice are quantified. (e–h) Cartilage degradation in medial regions of stifle joints from C57BL/6J mice subjected to DMM surgery and then treated by oral gavage with vehicle ($n = 8$) or imatinib 100 mg/kg/d ($n = 6$) for 12 weeks. Representative Safranin-O stained medial stifle joint sections from these mice are shown (e); arrowheads show severe cartilage loss. Cartilage degeneration (f), osteophyte formation (g), and synovitis (h) in medial regions of stifle joints from these mice are quantified. Symbols represent scores from individual mice. Bars denote mean ± s.d. *$p \leq 0.05$, **$p \leq 0.01$, by multiple comparisons one-way ANOVA. Scale bars, 200 µm. Scoring of joint pathologies was done by an investigator blinded to the experimental groups. Results are representative of three independent experiments for imatinib treatment, and two independent experiments for $Kit^{W-sh/W-sh}$ deficient mice. PBS, phosphate-buffered saline; BMCMCs, bone marrow-derived cultured mast cells; DMM, destabilization of the medical meniscus.

DOI: https://doi.org/10.7554/eLife.39905.006

The following figure supplements are available for figure 2:

**Figure supplement 1.** Genetic elimination of mast cells in a c-kit independent mouse model attenuates osteoarthritic development and severity.

DOI: https://doi.org/10.7554/eLife.39905.007

**Figure supplement 2.** Genetic deficiency of mast cells reduces synovitis and osteophyte formation following DMM.

DOI: https://doi.org/10.7554/eLife.39905.008

**Figure supplement 3.** Staining of mast cells in the synovium of mast cell-deficient and mast cell-engrafted mice following DMM.

DOI: https://doi.org/10.7554/eLife.39905.009

**Figure supplement 4.** Pharmacologic inhibition of mast cells by imatinib reduces synovitis and osteophyte formation following DMM.

DOI: https://doi.org/10.7554/eLife.39905.010

and *Cpa3-Cre;Mcl-1$^{+/+}$* control mice and the mast cell-engrafted mice (*Figure 2—figure supplement 3b and d*). Mast cell engraftment reversed the relative protection conferred by mast cell deficiency; that is, there was no overt difference in the degree of cartilage degradation (*Figure 2a and b*, *Figure 2—figure supplement 1a and b*), osteophyte formation (*Figure 2c*, *Figure 2—figure supplement 1c*, and *Figure 2—figure supplement 2*), or synovitis (*Figure 2d*, *Figure 2—figure supplement 1d*, and *Figure 2—figure supplement 2*) between mast cell-sufficient control mice and the corresponding mast cell-engrafted genetically mast cell-deficient mice 20 weeks after DMM. Given that several of the *Cpa3-Cre;Mcl-1$^{fl/fl}$* mice developed osteoarthritis (*Figure 2—figure supplement 1b*) and the *Cpa3-Cre;Mcl-1$^{fl/fl}$* mast cell deficiency is known to be incompletely penetrant with the presence of residual mast cells observed in certain organs and mice (*Lilla et al., 2011*; *Reber et al., 2012*; *Tsai et al., 2005*), we performed additional anti-tryptase immunostaining of the joint tissues from the *Cpa3-Cre;Mcl-1$^{fl/fl}$* mice that developed osteoarthritis following DMM to more comprehensively characterize these mice and their stifle joints for mast cell deficiency. In the *Cpa3-Cre;Mcl-1$^{fl/fl}$* mice that developed osteoarthritis, we observed peri-articular tryptase-positive mast cells suggesting that incomplete mast cell deficiency contributed to their development of osteoarthritis (*Figure 2—figure supplement 3e*). Together, these findings demonstrate that mast cells promote inflammation and cartilage damage in this mouse model of osteoarthritis.

To complement the genetic studies we determined whether pharmacological inhibition with imatinib mesylate (imatinib), a drug that potently inhibits several receptor tyrosine kinases, including c-kit (*Juurikivi et al., 2005*), a crucial factor for mast cell growth and survival, would be effective in limiting the development of osteoarthritis in wild-type mice. Compared with vehicle-treated mice, treatment with imatinib for 12 weeks following DMM significantly attenuated cartilage degradation (*Figure 2e and f*), osteophyte formation (*Figure 2g*, *Figure 2—figure supplement 4a*), and synovitis (*Figure 2h*, *Figure 2—figure supplement 4a*) associated with DMM-induced murine osteoarthritis. Furthermore, immunostaining with anti-tryptase revealed that the total number of mast cells in joints of imatinib-treated mice was significantly less than that in vehicle-treated mice (*Figure 2—figure supplement 4b and c*).

## Mast cell-derived tryptases promote osteoarthritis-associated pathology

Having established a pathogenic role for mast cells in osteoarthritis and because levels of the activated form of mast cell-derived tryptase are significantly elevated in the synovial fluids of individuals with osteoarthritis, a finding in agreement with previous reports (*Nakano et al., 2007*), we next investigated mechanisms by which tryptase might promote the pathogenesis of osteoarthritis. We first tested whether selectively inhibiting the protease activity of tryptase with APC366 (*Cairns, 2005*)

– an oral, selective tryptase small-molecule inhibitor previously used to alleviate allergic, inflammatory and fibrotic responses in multiple mouse models (*Lu et al., 2014*; *Matos et al., 2013*; *Sevigny et al., 2011*) - could effectively attenuate the progression and/or severity of osteoarthritis in mice. We found that following DMM, treatment with APC366 for 12 weeks significantly reduced cartilage damage (*Figure 3a and b*), osteophyte formation (*Figure 3c*, *Figure 3—figure supplement 1*) and synovitis (*Figure 3d*, *Figure 3—figure supplement 1*) compared to control mice treated with vehicle, suggesting that tryptase inhibition can prevent the development of osteoarthritis in mice. We, additionally, measured the expression of pro-inflammatory and degradative mediators known to be produced by mast cells in DMM joints following treatment with the tryptase inhibitor APC366. Six-weeks after DMM, transcriptional expression of multiple mediators including IL-1β, IL-6, IL-8, CCL2, CCL5, ADAMTS4 and MMP3 was significantly reduced in DMM synovial tissues derived from APC366-treated as compared to vehicle-treated mice (*Figure 3e*).

As tryptase has been shown to promote pathogenic properties in human rheumatoid arthritis-derived synovial fibroblasts (*Xue et al., 2012*), we examined whether tryptase could also induce pro-inflammatory and proliferative responses in primary synovial fibroblasts derived from remnant osteoarthritic joint tissue. Indeed, tryptase significantly increased the expression of the pro-inflammatory cytokine IL-1β and degradative enzymes MMP3 and ADAMTS4 (*Figure 3f*), increased the secretion of cytokines IL-1β (*Figure 3g*), IFNγ (*Figure 3h*), and increased synovial fibroblast proliferation in vitro, as demonstrated by increased expression of the activation marker Ki-67 by fibroblasts (*Figure 3i*). In vitro treatment of synovial fibroblasts with tryptase also promoted phosphorylation of Erk1/2, indicating that tryptase can activate pro-inflammatory signaling pathways in synovial fibroblasts (*Figure 3j and k*). Further, in vitro inhibition of tryptase activity with APC366 abrogated the pro-inflammatory and proliferative responses of synovial fibroblasts (*Figure 3f–i*).

## IgE deficiency attenuates osteoarthritis-associated pathology in mice

While mast cells can be activated by a wide range of stimuli, IgE mediates mast cell degranulation and release of biologically active mediators through cross-linking of the high affinity IgE receptor, FcεRI (*Galli and Tsai, 2012*; *Gilfillan and Tkaczyk, 2006*). We hypothesized that IgE might mediate mast cell activation in osteoarthritis. To determine the potential role of IgE in the pathogenesis of osteoarthritis, we subjected IgE-deficient ($Igh7^{-/-}$) mice and IgE-sufficient littermate controls ($Igh7^{+/+}$) to DMM. Twenty weeks after DMM surgery, IgE-deficient mice exhibited markedly diminished cartilage damage (*Figure 4a and b*), osteophyte formation (*Figure 4c*, *Figure 4—figure supplement 1a*), and synovitis (*Figure 4d*, *Figure 4—figure supplement 1a*).

To extend this observation, we treated mice with an anti-IgE neutralizing antibody that prevented IgE binding to FcεRI for 12 weeks following DMM surgery. Compared with isotype control-treated mice, treatment with anti-IgE antibody significantly attenuated cartilage degradation (*Figure 4e and f*), osteophyte formation (*Figure 4g*, *Figure 4—figure supplement 1b*), and synovitis (*Figure 4h*, *Figure 4—figure supplement 1b*). Together, these studies demonstrate that IgE plays a crucial role in promoting the pathogenesis of murine osteoarthritis.

## IgE signaling through FcεRI promotes pathogenesis of osteoarthritis

FcεRI, which is highly expressed on mast cells and basophils, is a tetrameric receptor comprising one α-chain that binds IgE, one β-chain that is a signal amplifier, and two γ-chains that initiate signaling via the spleen tyrosine kinase (Syk). To further define a role for FcεRI in the pathogenesis of osteoarthritis, we performed DMM surgeries in mice deficient in FcεRIα ($Fcer1a^{-/-}$). We found that mice deficient in FcεRIα, which as a consequence cannot transduce IgE signals, were significantly protected against osteoarthritic development as compared to wild-type controls (*Figure 5a*). Twenty weeks after DMM, FcεRIα-deficient mice developed significantly less cartilage damage (*Figure 5b*), osteophyte formation (*Figure 5c*, *Figure 5—figure supplement 1a*), and synovitis (*Figure 5d*, *Figure 5—figure supplement 1a*).

Given the central role for the tyrosine kinase Syk in FcεRI-mediated signaling, we evaluated whether pharmacologic inhibition of Syk using the potent and selective small molecule inhibitor PRT062607 (*Coffey et al., 2017*; *Spurgeon et al., 2013*) could ameliorate development of murine osteoarthritis. Treatment of mice with PRT062607 for 12 weeks following DMM markedly reduced the development and/or severity of osteoarthritis compared to vehicle-treated mice (*Figure 5e*).

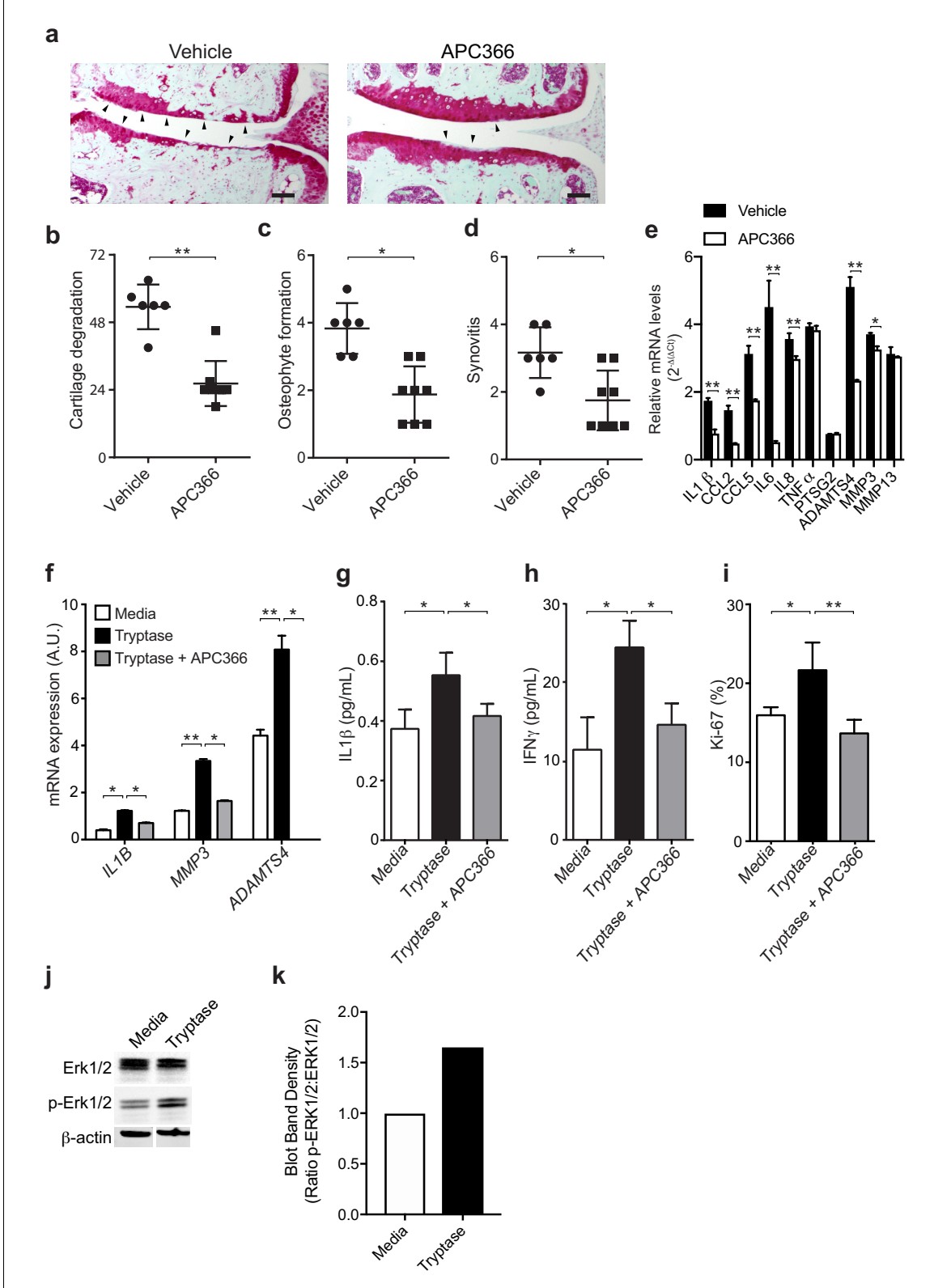

**Figure 3.** Mast cell-derived tryptases promote osteoarthritis pathology in vitro and in vivo. (a–e) Cartilage degradation in medial regions of stifle joints from C57BL/6J mice subjected to DMM surgery and then treated orally with the tryptase inhibitor APC366 5 mg/kg/d (n = 5) or vehicle (n = 7) for 12 weeks. Representative safranin-O stained medial stifle joint sections from these mice are shown (a); arrowheads show severe cartilage loss. Cartilage degeneration (b), osteophyte formation (c), and synovitis (d) in medial regions of stifle joints from these mice are quantified. Scoring of joint

*Figure 3 continued on next page*

*Figure 3 continued*

pathologies were done by two investigators blinded to experimental groups. Data are representative of three independent experiments. Symbols represent scores from individual mice. Bars are the mean ± s.d. for each group. *p≤0.05, **p≤0.01, by Mann Whitney test. Scale bars, 200 μm. (e) Relative mRNA expression of pro-inflammatory/degradative enzyme genes in mouse stifle joints. (f) Relative mRNA expression of inflammatory/degradative enzyme genes in osteoarthritic synovial fibroblasts treated for 24 hr with 0.2 μg/ml tryptase with or without 100 μM APC366. (g–h) Quantification of IL1β (g) and IFNγ (h) secretion by synovial fibroblasts stimulated for 24 hr. (i) Flow cytometric quantification of Ki-67 +synovial fibroblasts treated with media or 0.2 μg/ml tryptase with or without 100 μM APC366 for 72 hr. (j) Western blot analysis of total ERK1/2, phosphorylated ERK1/2 (p-ERK1/2), and β-actin in primary osteoarthritic synovial fibroblasts treated with media or 0.2 μg/ml tryptase for 72 hr. (k) Ratio of densitometry of p-ERK1/2:ERK1/2 bands from western blot in (j) Data in (f–i) are mean ± s.d. of triplicate values. *p≤0.05, **p≤0.01 by Student's t test. Results are representative of three independent experiments using samples from independent donors.

DOI: https://doi.org/10.7554/eLife.39905.011

The following figure supplement is available for figure 3:

**Figure supplement 1.** Representative images of osteophyte formation and synovitis in mice treated with the tryptase inhibitor APC366 following DMM.
DOI: https://doi.org/10.7554/eLife.39905.012

Inhibition of Syk by PRT062607 resulted in decreased cartilage damage (*Figure 5f*), osteophyte formation (*Figure 5g*, *Figure 5—figure supplement 1b*), and synovitis (*Figure 5h* and *Figure 5—figure supplement 1b*) relative to vehicle treatment. We used qPCR to analyze the levels of mRNAs encoding pro-inflammatory cytokines and degradative enzymes known to be produced by mast cells. Six-weeks following DMM, transcriptional expression of multiple pro-inflammatory cytokines and proteases including IL-1β, IL-6, CCL2, ADAMTS4 and MMP13 were significantly reduced in DMM joint tissues derived from mice treated with the Syk-inhibitor PRT062607 as compared to vehicle (*Figure 5i*). Together, these data suggest that the IgE/FcεRI/Syk axis mediates mast cell activation and degranulation and is a key pathogenic mechanism of osteoarthritis.

## Discussion

A major hurdle in the development of disease-modifying therapeutics for osteoarthritis is insufficient understanding of the cellular and molecular mechanisms underlying the pathogenesis of osteoarthritis. Mast cells have been implicated in the pathogenesis of various non-allergic, chronic, inflammatory diseases (*Theoharides et al., 2012*) including the etiologically-distinct inflammatory arthritides, including rheumatoid arthritis (*Lee et al., 2002*) and gouty arthritis (*Reber et al., 2014*). While it has been shown that mast cells and their mediators (e.g., histamine and tryptases) are present in osteoarthritic synovial tissue and fluids (*Buckley et al., 1998*; *Dean et al., 1993*; *Lee et al., 2013*; *Nakano et al., 2007*) and that their numbers in osteoarthritic synovial tissue correlate with structural damage in knee osteoarthritis (*de Lange-Brokaar et al., 2016*), their direct participation, mode of activation, and mechanisms by which they contribute to pathogenesis have not previously been shown. Here, we show that the activation and degranulation of mast cells via the IgE/FcεRI/Syk axis mediates inflammation and tissue damage in osteoarthritis, at least in part through mast cell-derived tryptase. Further, we demonstrate that pharmacologic interventions targeting mast cells at multiple levels reduce the severity of osteoarthritis in mice, including inhibition of (i) the mast cell growth factor receptor c-kit using imatinib (ii) mast cell-derived tryptases using APC366, (iii) IgE-mediated FcεRI-engagement through depletion of IgE, and (iv) FcεRI signaling by inhibiting its downstream signaling molecule Syk.

We demonstrate that the IgE/FcεRI/Syk signaling axis contributes to inflammation and cartilage damage in murine osteoarthritis. Given previous reports have shown that mast cells obtained from osteoarthritic synovial tissues express activating FcRs (e.g., FcγRI) as well as FcεRI at levels similar to those seen in rheumatoid arthritis (*Lee et al., 2013*), we propose a model wherein the IgE/FcεRI/Syk signaling axis in mast cells potentiates chronic inflammation in osteoarthritis.

While it is well documented that IgE is critical for initiating and sustaining chronic allergic inflammation, emerging evidence suggests that IgE-mediated cellular activation directly contributes to tissue remodeling (*Galli and Tsai, 2012*; *Roth et al., 2015*). Indeed, therapeutic strategies targeting IgE, such as omalizumab, reduce airway and tissue remodeling in allergic inflammatory conditions including as asthma and atopic dermatitis (*Oettgen, 2016*; *Strunk and Bloomberg, 2006*). Similar to the tissue remodeling in allergic diseases, human and murine osteoarthritis are characterized by

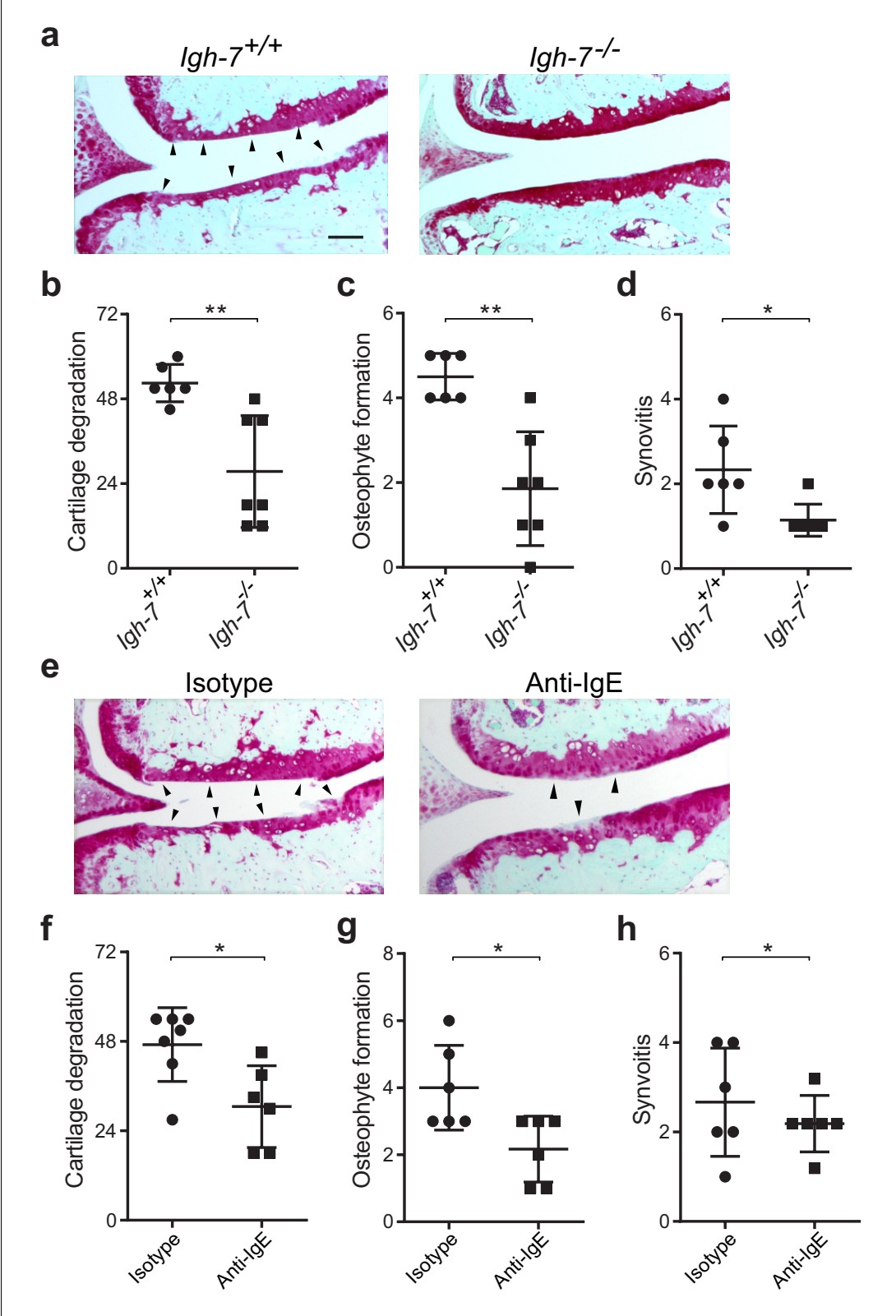

**Figure 4.** Genetic deficiency or pharmacologic depletion of IgE protects against the development of osteoarthritis in mice. (a–d) Cartilage degradation in medial regions of stifle joints from C57BL/6J IgE-deficient (*Igh7*[-/-], n = 7) and IgE-sufficient (*Igh7*[+/+], n = 6) mice 20 weeks after DMM surgery. Representative safranin-O stained medial stifle joint sections from these mice are shown (a); arrowheads show severe cartilage loss. Quantification of cartilage degradation (b), osteophyte formation (c), and synovitis (d). (e–h) Cartilage degradation in medial regions of stifle joints from C57BL/6J mice

*Figure 4 continued on next page*

**Figure 4 continued**

subjected to DMM surgery and then treated i.p. with anti-IgE antibody (*n* = 6) or isotype-matched control antibody (*n* = 7) 2.5 mg/kg twice per week for 12 weeks. Representative Safranin-O stained medial stifle joint sections from these mice are shown (**e**); arrowheads show severe cartilage loss. Cartilage degeneration (**f**), osteophyte formation (**g**), and synovitis (**h**) in medial regions of stifle joints from these mice are quantified. Symbols represent scores from individual mice. Bars denote mean ± s.d. *p≤0.05, **p≤0.01, by Mann Whitney test. Scale bars, 200 μm. Scoring of joint pathologies was performed by an investigator blinded to experimental groups. Data are representative of two independent experiments with similar results.

DOI: https://doi.org/10.7554/eLife.39905.013

The following figure supplements are available for figure 4:

**Figure supplement 1.** Deficiency of IgE reduces synovitis and osteophyte formation in mice subjected to DMM.

DOI: https://doi.org/10.7554/eLife.39905.014

**Figure supplement 2.** Mast cell numbers in DMM joint tissue in IgE deficient and wild-type mice following DMM.

DOI: https://doi.org/10.7554/eLife.39905.015

abnormal and progressive bone and other tissue remodeling (*Remst et al., 2014*) which result in altered joint biomechanics that further promote development of osteoarthritis. Phosphorylation of Syk is a critical downstream signaling event for the transmission of IgE/FcεRI signals. Here, we demonstrate that pharmacologic depletion of IgE, or blockade of IgE-mediated FcεRI signaling using a Syk inhibitor, reduce bone remodeling (as measured by osteophyte formation) and disease severity in the DMM murine model of osteoarthritis. Together, these findings strongly implicate IgE-mediated mast cell activation in bone and synovial tissue remodeling in osteoarthritis.

Previous reports found that IgE levels are not elevated in the synovial fluids of osteoarthritic joints (*Hunder and Gleich, 1974*), and in preliminary studies we observed only trends towards an association of increased total IgE in serum with osteoarthritis in humans. It remains possible that antigen-specific IgE contributes to mast cell activation in osteoarthritis, and future studies will be needed to further investigate the relative contributions of antigen-specific IgE-dependent and other mechanisms of mast cell activation in the pathogenesis of osteoarthritis.

While many classical IgE-mediated allergic diseases including asthma, allergic rhinitis and eczema exhibit comorbidities (*Pedersen and Weeke, 1983*), we are not aware of evidence of a clear link between osteoarthritis and allergic diseases. There is significant evidence that these classic allergic diseases are caused by antigen-specific IgE-dependent activation of mast cells (*Galli and Tsai, 2012*; *Hamelmann et al., 1997*; *Oettgen and Geha, 2001*; *van der Heijden et al., 1993*). If the role of mast cells in osteoarthritis pathogenesis is dependent on antigen-specific IgE, the target antigens could potentially be exogenous allergens. Another possibility is that in osteoarthritis the IgE target antigens are bone or cartilage breakdown products that give rise to neoantigens following joint injury or instability. Although we are not aware of evidence that osteoarthritis is associated with classic allergic and/or IgE-dependent diseases, an epidemiological analysis that addresses this important question is warranted. Antigen-specific antibody responses can also be generated in autoimmune responses, however osteoarthritic synovial linings do not exhibit histologic features consistent with an adaptive autoimmune response and we do not believe our findings suggest the presence of an classical adaptive autoimmune response. Future studies are needed to determine if IgE targets specific antigens, and to further characterize the role of IgE in osteoarthritis.

Mechanical instability and stresses likely contribute to the pathogenesis of osteoarthritis in a significant subset of patients. We do not believe that a role for mechanical stresses is inconsistent with a role for IgE and mast cells, and it is possible that mechanical stresses produce cartilage breakdown products and/or cellular responses that promote IgE-dependent mast cell activation.

In addition to IgE-mediated activation of mast cells, a wide range of physical, biological, and chemical triggers can contribute to mast cell activation, including products of complement activation, platelet-activating factor, damage-associated molecular patterns (DAMPs), and a number of endogenous peptides (e.g., vasoactive intestinal polypeptide [VIP], and substance P [SP]) (*Galli and Tsai, 2012*; *Gaudenzio et al., 2016*). We previously demonstrated that complement plays a critical role in the pathogenesis of osteoarthritis (*Lepus et al., 2014*; *Wang et al., 2011*). Activation of the complement system results in the production of C3a and C5a which serve as anaphylatoxins and activators of mast cells (*Erdei et al., 2004*; *Gaudenzio et al., 2016*). It is therefore possible that complement, in addition to Fc(ε)RI, regulates the recruitment and activation of mast cells in the synovium to promote the pathogenesis of osteoarthritis. We previously demonstrated that the cartilage

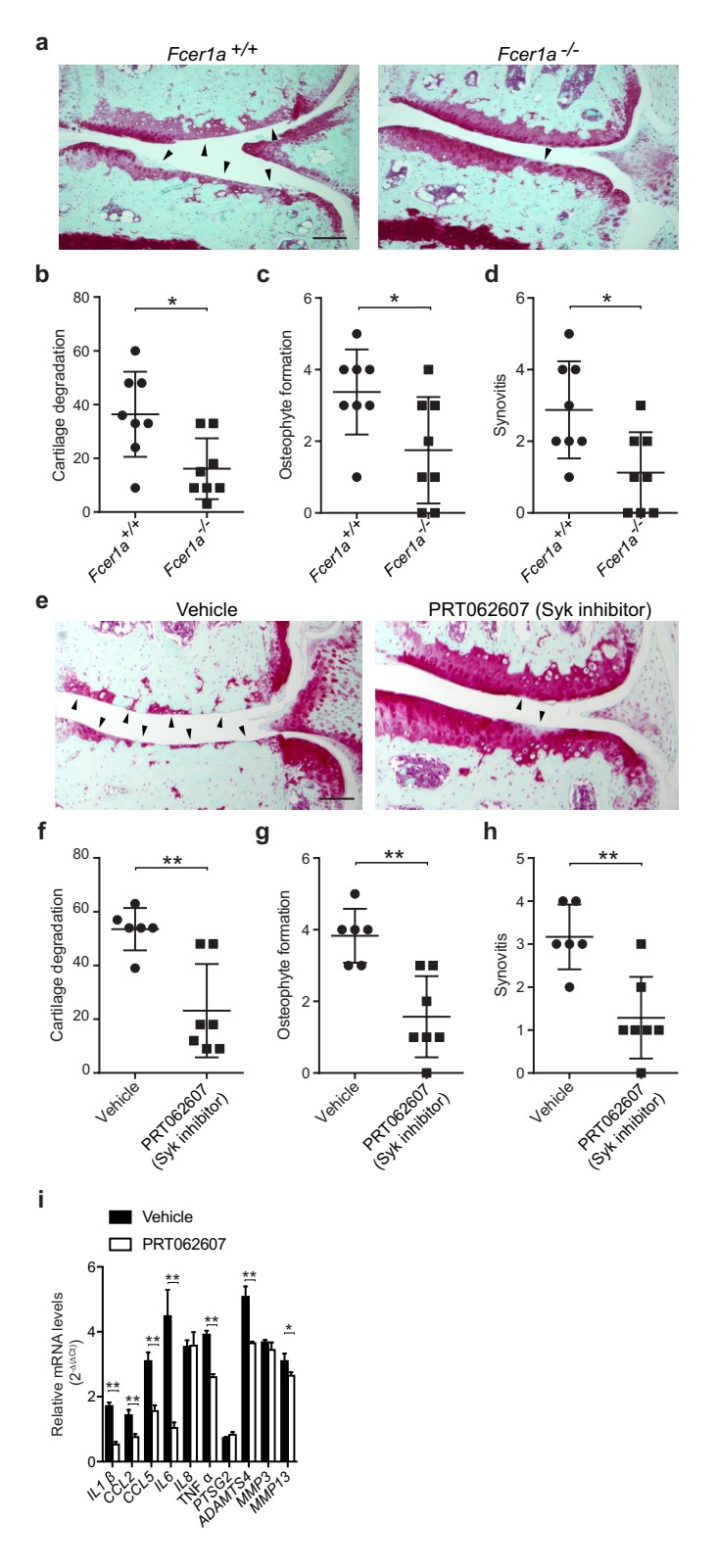

**Figure 5.** A critical role for IgE-mediated signaling through FcεRI and Syk in osteoarthritis. (**a–d**) Cartilage degradation in medial regions of stifle joints from C57BL/6J FcεRIα-sufficient (*Fcer1a⁺/⁺·n = 8*) and FcεRIα-deficient (*Fcer1a⁻/⁻·n = 8*) mice 20 weeks after DMM surgery. Representative safranin-O stained medial stifle joint sections from these mice are shown; arrowheads show severe cartilage loss. Cartilage degradation (**b**), osteophyte formation (**c**), and synovitis (**d**) in medial regions of stifle joints from these mice are quantified. Symbols represent scores from individual mice. Bars denote
*Figure 5 continued on next page*

*Figure 5 continued*

mean ± s.d. *p≤0.05, **p≤0.01, by Mann Whitney test. Scale bars, 200 µm. Scoring of joint pathologies was done by two investigators blinded to experimental groups. Data are representative of two independent experiments with similar results. DMM, destabilization of the medical meniscus. (e–h) Cartilage degradation in medial regions of stifle joints from C57BL/6J mice subjected to DMM surgery and then orally with vehicle (n = 6) or 75 mg/Kg/ day of the Syk inhibitor PRT062607 (n = 7), for 12 weeks. Representative Safranin-O stained medial stifle joint sections from these mice are shown (e); arrowheads show severe cartilage loss. Cartilage degeneration (f), osteophyte formation (g), and synovitis (h) in medial regions of stifle joints from these mice are quantified. Symbols represent scores from individual mice. Bars denote mean ± s.d., **p≤0.01, by Mann Whitney test. Scale bars, 200 µm. Scoring of joint pathologies was done by two investigators blinded to experimental groups. (i) Relative mRNA expression of pro-inflammatory/ degradative enzyme genes in mouse stifle joints. Data are representative of two independent experiments with similar results. DMM, destabilization of the medical meniscus.

DOI: https://doi.org/10.7554/eLife.39905.016

The following figure supplement is available for figure 5:

**Figure supplement 1.** Deficiency in *FceR1a* or blockade of FcεRIα signaling in mice reduces synovitis and osteophyte formation following DMM.

DOI: https://doi.org/10.7554/eLife.39905.017

breakdown product, fibromodulin, can activate the complement system (*Wang et al., 2011*), and it is possible that mast cell tryptase-mediated cartilage degeneration produces fibromodulin and other cartilage breakdown products that reciprocally activate complement in the synovia in osteoarthritis. It will be important to define the temporal relationship between mast cell activation/degranulation and complement activation in the development and progression of osteoarthritis, including how activation of one affects the other and vice versa. Further, complement is capable of stimulating mast cells alone, and can also enhance IgE-dependent mast cell activation and degranulation (*Schäfer et al., 2013*). Future studies will be needed to further define the roles and relationships of complement activation, IgE, and mast cell activation/degranulation in the development and progression of osteoarthritis.

The development and survival of mast cells is critically dependent on signaling via the stem cell factor (SCF) receptor c-kit (CD117), a member of the receptor tyrosine kinase family. While c-kit is widely expressed by hematopoietic progenitor and germ cells, in the context of the mature immune system only mast cells robustly express c-kit, and in the synovial compartment c-kit is predominantly expressed by mast cells (*Ceponis et al., 1998*; *Galli et al., 1993*). Imatinib mesylate, an orally available small molecule with potent and selective inhibitory activity against several tyrosine kinases, including Abl, c-kit and platelet-derived growth factor receptor, has been shown to inhibit pro-inflammatory cytokine production from mast cells (*Paniagua et al., 2006*) and to promote apoptosis of mast cells (*Juurikivi et al., 2005*). Furthermore, long-term treatment with imatinib induced severe mast cell deficiency and reduced serum tryptase levels without inducing adverse effects in patients with chronic myeloid leukemia (*Cerny-Reiterer et al., 2015*), suggesting that imatinib suppresses mast cell production and/or survival. Treatment of severe asthmatics with imatinib for 6 months also reduced mast cell numbers, bronchial hyperresponsiveness, and tryptase levels (*Cahill et al., 2017*). Further, it has been previously reported that imatinib reduces the development and severity of arthritis in two distinct mouse models of rheumatoid arthritis (*Koyama et al., 2007*; *Paniagua et al., 2006*). Here, we show that treatment with imatinib, reduces numbers of synovial tryptase-expressing mast cells in mouse knees and attenuates murine osteoarthritis. Indeed, imatinib may play a broader role in mitigating the pathogenesis of osteoarthritis as it can influence pro-inflammatory responses via receptor tyrosine kinase inhibition on cell types including macrophages, B cells, and T cells (*Paniagua et al., 2006*; *Zitvogel et al., 2016*). However, as shown here, attenuation of mast cell responses is likely responsible, at least in part, for the effectiveness of imatinib in preventing the development of osteoarthritis in mice.

Mast cells could participate in osteoarthritis through multiple mechanisms, including release of pre-formed mediators such as tryptases and de novo synthesis of cytokines that could further propagate inflammation, promoting a vicious cycle of inflammation and tissue damage. Because we found high levels of active tryptase in osteoarthritic synovial fluids and previous studies have shown tryptases can induce cartilage aggrecanolysis in vitro (*Magarinos et al., 2013*), and promote inflammatory responses in mouse models of autoimmune arthritis (*McNeil et al., 2008*; *Shin et al., 2009*), we investigated the mechanisms by which tryptases could directly influence osteoarthritis-associated pathogenic processes. Our data suggest that mast cell-derived tryptases promote cartilage

degeneration, support synovial fibroblast proliferation and release of pro-inflammatory and degradative mediators from joint tissues. Importantly, we demonstrate that inhibition of tryptase activity using the tryptase-specific inhibitor APC366 attenuates osteoarthritis in mice. Tryptases are natural agonists of proteinase activated receptor-2 (PAR2), which is expressed by a wide variety of cells including osteoarthritis synovial fibroblasts. Importantly, PAR2 deficiency significantly attenuates the development and severity of osteoarthritis in mice (*Huesa et al., 2016*; *Jackson et al., 2014*). Therefore, it is conceivable but remains to be formally tested that tryptase signaling via PAR2 promotes inflammatory and degradative responses in osteoarthritis.

Although we have not formally demonstrated that mast cells are the source of tryptases that promote the pathogenesis of osteoarthritis, our data demonstrate that: (i) mast cells comprise 1–3% of synovial cells in synovial linings from both osteoarthritic joints and joints with prior injury but no radiographic osteoarthritis, (ii) mast cells are actively degranulating to release tryptase in human osteoarthritic synovial linings, and (iii) pharmacologic inhibition of typtase prevents the development of osteoarthritis in mice. However, mast cells are the predominant producer of tryptases, and to a much lesser extent basophils (*Schwartz, 2006*). Together, these data suggest that mast cells are the predominant cellular source of tryptase in human and murine osteoarthritis.

There are multiple potential limitations to this study. First, it was previously shown that mast cells are associated with joint pain in murine osteoarthritis (*Sousa-Valente et al., 2018*) and that synovial mast cell numbers are associated with the degree of synovitis in human osteoarthritis (*de Lange-Brokaar et al., 2016*). We and others previously demonstrated that following DMM in mice development of histologic osteoarthritis is associated with poor functional outcomes including pain and abnormal gait (*Huesa et al., 2016*; *Wang et al., 2011*). The studies presented in this manuscript demonstrate that mast cells and dysregulated mast cell activation contribute to cartilage and joint degeneration following DMM, and based on this prior work (*Huesa et al., 2016*; *Wang et al., 2011*) such pathologic changes are anticipated to result in pain and abnormal gait. Future studies will be needed to further characterize the role of mast cells in osteoarthritis-associated pain and joint dysfunction. Second, while our findings demonstrate a critical role for mast cells, mast cell activation pathways, and the mast cell product tryptase in the development of osteoarthritis following DMM, further investigation is needed to characterize the relationship between IgE-mediated mast cell activation and the presence of pro-inflammatory cytokines and proteases implicated in synovitis and cartilage degradation in osteoarthritis. Additionally, demonstration in vivo that IgE/Syk signaling is associated with increased tryptase production would further strengthen our findings.

Together, our results demonstrate that IgE/FcεRI/Syk axis-activated mast cells promote the development of osteoarthritis following mechanical injury (DMM) in mice. As the DMM model is most representative of osteoarthritis development following traumatic joint injury (PTOA) in humans, these findings suggest that PTOA arises as result of activation of the IgE/FcεRI/Syk axis. It is possible that non-traumatic osteoarthritis, calcium pyrophosphate crystal-associated osteoarthritis, or other subsets of osteoarthritis may arise from activation of other molecular pathways.

We propose a model wherein IgE-mediated mast cell activation via FcεRI and Syk which results in mast cell degranulation and the release of pro-inflammatory and degradative mediators, including tryptase, leads to cartilage and joint breakdown. This results in a vicious cycle of tissue damage, inflammation, and unchecked mast cell activation, and thus causes the development and progression of osteoarthritis. Our findings demonstrate a central role for IgE-mediated mast cell activation in the pathogenesis of osteoarthritis, and provide the rationale for targeting mast cells or tryptase as a disease-modifying therapeutic strategy for osteoarthritis.

## Materials and methods

### Key resources table

| Reagent type (species) or resource | Designation | Source or reference | Identifiers | Additional information |
|---|---|---|---|---|
| Genetic reagent (M. musculus) | Kit[W-sh/W-sh] | The Jackson Laboratory | Stock No. 012861 | |

*Continued on next page*

*Continued*

| Reagent type (species) or resource | Designation | Source or reference | Identifiers | Additional information |
|---|---|---|---|---|
| Genetic reagent (M. musculus) | Cpa3-Cre;Mcl-1$^{fl/fl}$ | *Lilla et al., 2011* | | Dr. Stephen Galli (Stanford University) |
| Genetic reagent (M. musculus) | Cpa3-Cre;Mcl-1$^{+/+}$ | *Lilla et al., 2011* | | Dr. Stephen Galli (Stanford University) |
| Genetic reagent (M. musculus) | Fcer1a$^{-/-}$ | The Jackson Laboratory | Stock No.10512 | |
| Genetic reagent (M. musculus) | C57BL/6J | The Jackson Laboratory | Stock No. 000664 | |
| Antibody | Anti-Mast Cell Tryptase antibody | Abcam | catalog #:ab2378 clone: AA1 | |
| Antibody | Mouse IgG1, kappa Isotype Control | Crown Biosciences | catalog #: c0005 | |
| Antibody | Phospho-p44/42 MAPK (Erk1/2) (Thr202/Tyr204) Antibody | Cell signaling | catalog #: 9101 | |
| Antibody | p44/42 MAPK (Erk1/2) Antibody | Cell signaling | catalog #: 4695 | |
| Antibody | Anti-beta actin antibody | Abcam | catalog #: Ab8227 | |
| Commercial assay or kit | Mast Cell Degranulation Assay Kit, | Millipore | catalog #: IMM001 | |
| Chemical compound, drug | imatinib mesylate | LC Laboratories | catalog #: I-5508 | |
| Chemical compound, drug | APC366 | Tocris | catalog #: 2511 | |
| Chemical compound, drug | PRT062607 | Synnovator | catalog #: 1370261-97-4 | |

### Human samples

All human samples were obtained and studied under protocols that included written informed consent and consent to publish and that were approved by the Stanford University Institutional Review Board (IRB) (approval #3780) and the University of Padova IRB (approval #39872). Osteoarthritic synovial membranes were obtained at the time of total joint replacement from individuals with end-stage osteoarthritis at the VA Palo Alto Health Care System. Synovial fluids were obtained from individuals with varying degrees of osteoarthritis severity as assessed by K-L score. Synovial membranes and synovial fluids from individuals undergoing arthroscopic anterior cruciate ligament reconstruction surgery who had no arthroscopic evidence of articular cartilage loss were used as controls.

### Transmission electron microscopy (TEM) analysis of human synovium

Synovial membranes from five osteoarthritic knees and five non-osteoarthritic control knees with prior joint trauma >6 months ago but no radiographic osteoarthritis were analyzed, and after staining with immuno-gold labeled anti-tryptase (Abcam, clone AA1) or isotype-matched control antibody the entire post-etch-embedded section for each sample was scanned by TEM. The number of mast cells and degranulated mast cells was determined by an examiner blinded to the experimental group of each sample. Mast cells were identified based on the presence of electron dense granules containing gold-labelled tryptase particles. Degranulation was determined by assessment of fusion of granule plasma membranes, fusion of granule membranes with cell membranes, exteriorization of granules, and presence of tryptase particles outside the exteriorized granule. The percent of degranulating mast cells per total mast cells was calculated for each sample. Multiple independent experiments were performed, and representative images and results are presented.

## Detailed methods

Synovial lining tissue was fixed in Karnovsky's fixative containing 2% glutaraldehyde and 4% paraformaldehyde in 0.1 M sodium cacodylate. After an initial ~30 min fixation, the specimens were cut into ~1 mm$^3$ pieces and returned to fresh fixative for 16–24 hr at 4°C. The specimens were washed with 100 mM cacodylate buffer, fixed with 1% osmium tetroxide for 1 hr, washed with excess distilled water then en bloc stained with 1% aqueous uranyl acetate overnight at 4°C. Samples were then dehydrated in a series of ethanol washes, propylene oxide, and embedded in resin. We picked up 75–90 nm sections on formvar/Carbon-coated slot Cu grids, stained them for 30 s in 3.5% uranyl acetate in 50% acetone followed by staining them in 0.2% lead citrate for 3 min. Post-embedding immunolabelling was carried out by micro etching in 10% periodic acid, followed by treatment with 10% sodium meta-periodate. Sections were blocked and stained with anti-human mast cell tryptase antibody (Abcam, clone AA1), or an isotype-matched IgG1 control antibody (Abcam), followed by incubation with a goat anti-mouse antibody conjugated with 10 nm Gold particles (British Biocell). Sections were then observed in a JEOL JEM-1400 120kV transmission electron microscope (JEOL USA) and images captured using a Gatan Orius 4k × 4 k digital camera.

## Measurement of active tryptase in synovial fluids

Levels of active tryptase were measured with the tosyl-gly-pro-lys-pNA substrate assay (Mast Cell Degranulation Assay Kit, Millipore) according to the manufacturer's protocols.

## Analysis of mast cell-related gene expression

We downloaded publicly available data from the US National Center for Biotechnology Information Gene Expression Omnibus (NCBI GEO accession codes GSE32317) comparing gene expression profiles of synovial membranes obtained from patients with early- or end-stage osteoarthritis and from individuals with prior joint trauma >6 months ago but no radiographic osteoarthritis (annotated as 'healthy' in the online dataset). All microarray analyses were restricted to putative mast cell- and mast cell activation-related genes. Unsupervised and supervised hierarchical clustering analyses were performed on the microarray data by using Cluster and TreeView software. Significance Analysis of Microarrays (SAM) analyses were used for determining statistical significance with a q-value cutoff set at 0.05. Paired or unpaired student's t-tests were employed where appropriate and p<0.05 was considered statistically significant.

## Surgical induction of osteoarthritis in mice

This study was performed in accordance with the recommendations in the Guide for the Care and Use of Laboratory Animals of the National Institutes of Health. All mouse studies were performed under protocols approved by the Stanford University Administrative Panel on Laboratory Animal Care (APLAC approval # 9942) and VA Palo Alto Health Care System Institutional Animal Care and Use Committees (IACUC approvals #ROW1552 and #ROW1755). Littermate controls were used for *Cpa3-Cre;Mcl-1*$^{fl/fl}$ (B6-*Cpa3-Cre;Mcl-1*$^{+/+}$), *Igh7*$^{-/-}$ (*Igh7*$^{+/+}$), and *Fcer1a*$^{-/-}$ (*Fcer1a*$^{+/+}$). A fully congenic *Kit*$^{W-sh/W-sh}$ mouse strain on a C57BL6/J genetic background (Stock No. 012861) and age-matched C57BL/6J (Stock No. 000664) were obtained from The Jackson Laboratory. Destabilization of the medial meniscus (DMM) was performed as described previously (*Glasson et al., 2007*; *Raghu et al., 2017*). Five to eight mice were used per experimental arm based on power calculations performed using the PS Power and Sample Size Calculations software program (W.D. Dupont and W.D. Plummer, Department of Biostatistics, Vanderbilt University; Version 2.1.3.0).

## Mast cell engraftment studies

To generate mast cell-engrafted mice, we injected 4-week-old, male, mast-cell deficient *Kit*$^{W-sh/W-sh}$ mice and *Cpa3-Cre;Mcl-1*$^{fl/fl}$ (Hello *Kit*ty) mice intravenously (i.v.) with 10$^7$ wild-type bone marrow-derived cultured mast cells (BMCMCs; generated as previously described [*Grimbaldeston et al., 2005*]), and 8 weeks later with 10$^6$ BMCMCs intra-articularly (i.a.) into the stifle joints. Age-matched mast cell-deficient littermate mice injected both i.v. and i.a. with PBS (*Kit*$^{W-sh/W-sh}$ + PBS or *Cpa3-Cre;Mcl-1*$^{fl/fl}$ +PBS) and mast cell-sufficient mice injected both i.v. and i.a. with PBS (C57BL/6J + PBS or B6-*Cpa3-Cre;Mcl-1*$^{+/+}$ + PBS) were used as controls. DMM was then performed at 16 weeks of

age. Mast cell engraftment was assessed by toluidine blue staining of stifle joint sections from mice sacrificed 20 weeks after DMM.

## Pharmacologic treatment of murine osteoarthritis

Twenty-week-old wild-type C57BL/6J mice were randomized by cage to receive vehicle (water), 100 mg/kg/day imatinib mesylate (LC laboratories) or 5 mg/kg/day APC366 (Tocris) divided between two daily doses or 75 mg/Kg/day PRT062607 (synnovator) once daily by oral gavage for 12 weeks (beginning 24 hr after DMM surgery). Two of eight mice treated with 100 mg/kg/day imatinib were excluded from analyses due to inadequate histology. Similarly, for anti-IgE treatment, twenty-week-old wild-type C57BL/6J mice were administered 2.5 mg/Kg mouse anti-IgE antibody or IgG1κ isotype (Crown Biosciences) i.p. twice a week for 12 weeks. Mouse anti-mouse IgE was made using the sequences derived from the rat hybridoma (R1E4) that specifically binds to the region of mouse IgE known to bind FcεRI (*Ota et al., 2009*). Mice were sacrificed 12 weeks after DMM for histologic assessment of osteoarthritic development.

## Histologic assessment of osteoarthritic development in mice

Stifle joints were harvested 12 or 20 weeks after DMM and fixed in 10% neutral buffered formalin followed by decalcification in formic acid for 48 hr. Joints were then embedded in paraffin, and 6 µm sections cut from three separate levels of the joint and stained with Safranin-O for assessment of cartilage damage; H and E for assessment of synovial thickening (synovitis) and osteophyte formation; and toluidine blue for the assessment of mast cells. Cartilage degeneration, synovitis, and osteophyte formation were evaluated by two blinded observers using a modified version of a described scoring system (*Kamekura et al., 2005*) as we previously described (*Wang et al., 2011*). In brief: Cartilage degeneration was calculated by depth of cartilage degeneration (score of 0–4)× width of cartilage degeneration (with a score of 1 meaning one-third of the surface area, a score of 2 meaning two-thirds of the surface area, and a score of 3 meaning the whole surface area) in each third of the femoral-medial and tibial-medial condyles. The scores for the six regions were then summed. Synovitis scores were calculated as previously described (*Blom et al., 2004*): 0, no changes compared to normal joints; 1, thickening of the synovial lining and some influx of inflammatory cells; 2, thickening of the synovial lining and intermediate influx of inflammatory cells; and 3, profound thickening of the synovial lining (more than four cell layers) and maximal observed influx of inflammatory cells. Scores for synovitis were recorded for the femoral-medial and the tibial-medial condyles, and the scores for the two regions summed. Osteophyte formation was scored according to a previously described scoring system (*Kamekura et al., 2005*): 0, none; 1, formation of cartilage-like tissues; 2, increase of cartilaginous matrix; 3, endochondral ossification. Mast cells were quantified by a blinded examiner who determined the number of toluidine blue-positive mast cells per high power field of the joint sections.

## Immunohistochemical staining of murine joint sections for tryptase

Synovial sections were fixed, decalcified, blocked, and stained with a biotinylated anti-tryptase antibody (Abcam, clone AA1), followed by avidin-HRP, then TMB substrate, and microscopy performed to determine if tryptase-positive mast cells were present.

## In vitro tryptase stimulation assays

Primary synovial fibroblasts were derived from synovium of individuals with end-stage osteoarthritis by enzymatic digestion with 2 mg/ml Collagenase Type IV for 24 hr at 37°C. Passage 3 (P3) fibroblasts were serum starved overnight in 1% fetal bovine serum and then stimulated with media alone (Alpha MEM) or with 0.2 µg/ml tryptase in the presence or absence of 100 µM of the tryptase-selective inhibitor APC366. Their mRNA was isolated, and mRNA levels of pro-inflammatory mediators were measured by qPCR and normalized to those of 18 s. Taqman probes were obtained from Applied Biosystems. Proinflammatory cytokine and chemokine secretion was measured by multiplexed, fluorescent bead-based immunoassay (Luminex) by using the human cytokine 27-plex assay (Bio-Rad). For analysis of Erk activation, $10^3$ primary synovial fibroblasts were stimulated with media alone (Alpha MEM) or with 1 µg/ml tryptase for 30 min and then lysates were run on SDS PAGE gel

and western blot analysis was performed with anti-Erk1/2, anti-phosphoErk1/2 or anti-β-actin antibodies.

## Statistics

For analyses involving a single comparison, statistical comparisons were performed using either a two-tailed Student's *t* test or Mann-Whitney *U* test following tests for variance homogeneity. Multiple comparisons were performed using a one-way analysis of variance (ANOVA) followed by Dunnett's post-hoc test.

## Study approval

All human samples were obtained under protocols approved by the Stanford Institutional Review Board (IRB) or the University of Padova IRB, and written informed consent was received from participants prior to inclusion in the study. Participants were identified by numbers, never by name, in this study. All mouse breeding and osteoarthritis studies were performed under protocols approved by the Stanford Committee of Animal Research and in accordance with National Institutes of Health guidelines.

# Acknowledgements

These studies were supported by: VA RR&D Merit Review Awards I01BX002345, I01RX000934, I01RX000588 and I01RX002689 to WHR. Studies at the Cell Science Imaging Facility at Stanford were supported in part by ARRA Award Number 1S10RR026780-01 from the National Center for Research Resources (NCRR). Studies were supported by NIH/NIAMS grant R01AR067145 and NIH/NIAID grant R01AI132494 to SJG. The authors would like to thank Mariola Leiberbasch and Phillip Starkl for their technical assistance, John Perrino at the Cell Science Imaging Facility at Stanford for technical guidance and performance of the electron microscopy studies.

# Additional information

## Competing interests

Lawrence B Schwartz: Virginia Commonwealth University collects royalties for the tryptase assay licensed to Thermofisher and for tryptase antibodies licensed to Millipore and Santa Cruz, which are shared with Lawrence B Schwartz, who also is a paid consultant for Genentech. The other authors declare that no competing interests exist.

## Funding

| Funder | Grant reference number | Author |
|---|---|---|
| U.S. Department of Veterans Affairs | I01BX002345 | Qian Wang<br>Christin M Lepus<br>Harini Raghu<br>Heidi H Wong<br>Ericka von Kaeppler<br>Nithya Lingampalli<br>Michelle S Bloom<br>Jeremy Sokolove<br>William H Robinson |
| U.S. Department of Veterans Affairs | I01RX000588 | Qian Wang<br>Christin M Lepus<br>Harini Raghu<br>Heidi H Wong<br>Ericka von Kaeppler<br>Nithya Lingampalli<br>Michelle S Bloom<br>Nick Hu<br>Eileen E Elliott<br>Jeremy Sokolove<br>William H Robinson |

| U.S. Department of Veterans Affairs | I01RX000934 | Qian Wang<br>Christin M Lepus<br>Harini Raghu<br>Heidi H Wong<br>Ericka von Kaeppler<br>Nithya Lingampalli<br>Michelle S Bloom<br>Nick Hu<br>Eileen E Elliott<br>Jeremy Sokolove<br>William H Robinson |
|---|---|---|
| U.S. Department of Veterans Affairs | I01RX002689 | Qian Wang<br>Christin M Lepus<br>Harini Raghu<br>Heidi H Wong<br>Ericka von Kaeppler<br>Nithya Lingampalli<br>Michelle S Bloom<br>Nick Hu<br>Eileen E Elliott<br>Jeremy Sokolove<br>William H Robinson |
| National Institutes of Health | R01AR067145 | Laurent L Reber<br>Mindy M Tsai<br>Stephen J Galli |
| National Institutes of Health | R01AI132494 | Laurent L Reber<br>Mindy M Tsai<br>Stephen J Galli |
| National Center for Research Resources | 1S10RR026780-01 | William H Robinson |

The funders had no role in study design, data collection and interpretation, or the decision to submit the work for publication.

### Author contributions

Qian Wang, Conceptualization, Formal analysis, Funding acquisition, Investigation, Visualization, Methodology, Writing—original draft, Project administration, Writing—review and editing; Christin M Lepus, Conceptualization, Formal analysis, Investigation, Visualization, Methodology, Writing—original draft; Harini Raghu, Conceptualization, Data curation, Formal analysis, Investigation, Writing—original draft; Laurent L Reber, Data curation, Formal analysis, Investigation, Writing—original draft; Mindy M Tsai, Heidi H Wong, Formal analysis, Investigation, Reviewed and approved the manuscript; Ericka von Kaeppler, Data curation, Formal analysis, Investigation, Reviewed and approved the manuscript; Nithya Lingampalli, Michelle S Bloom, Data curation, Formal analysis, Investigation, Writing—original draft, Writing—review and editing; Nick Hu, Eileen E Elliott, Investigation, Reviewed and approved the manuscript; Francesca Oliviero, Resources, Investigation, Writing—original draft, Writing—review and editing; Leonardo Punzi, Nicholas J Giori, Stuart B Goodman, Resources, Investigation, Reviewed and approved the manuscript; Constance R Chu, Jeremy Sokolove, Resources, Formal analysis, Reviewed and approved the manuscript; Yoshihiro Fukuoka, Lawrence B Schwartz, Resources, Formal analysis, Investigation, Reviewed and approved the manuscript; Stephen J Galli, Conceptualization, Formal analysis, Methodology, Funding Acquisition, Reviewed and approved the manuscript; William H Robinson, Conceptualization, Resources, Formal analysis, Funding acquisition, Investigation, Methodology, Writing—original draft, Project administration, Writing—review and editing

### Author ORCIDs

William H Robinson  http://orcid.org/0000-0003-4385-704X

### Ethics

Human subjects: All human samples were obtained and studied under protocols that included written informed consent and consent to publish and that were approved by the Stanford University Institutional Review Board (IRB) (approval #3780) or the University of Padova IRB (approval #39872).

Animal experimentation: This study was performed in accordance with the recommendations in the Guide for the Care and Use of Laboratory Animals of the National Institutes of Health. All mouse studies were performed under protocols approved by the Stanford University Administrative Panel on Laboratory Animal Care (APLAC approval # 9942) and VA Palo Alto Health Care System Institutional Animal Care and Use Committees (IACUC approvals #ROW1552 and #ROW1755).

## Decision letter and Author response
Decision letter https://doi.org/10.7554/eLife.39905.022
Author response https://doi.org/10.7554/eLife.39905.023

## Additional files

### Supplementary files
• Transparent reporting form
DOI: https://doi.org/10.7554/eLife.39905.018

### Data availability
Expression data is available in the Gene Expression Omnibus (GEO) under accession number GSE32317.

The following previously published dataset was used:

| Author(s) | Year | Dataset title | Dataset URL | Database and Identifier |
|---|---|---|---|---|
| Wang Q, Rozelle AL, Lepus CM, Scanzello CR, Song JJ, Larsen DM, Crish JF, Bebek G, Ritter SY, Lindstrom TM, Hwang I, Wong HH, Punzi L, Encarnacion A, Shamloo M, Goodman SB, Wyss-Coray T, Goldring SR, Banda NK, Thurman JM, Gobezie R, Crow MK, Holers VM, Lee DM | 2011 | Gene expression in synovial membranes from patients with early and end-stage osteoarthritis | https://www.ncbi.nlm.nih.gov/geo/query/acc.cgi?acc=GSE32317 | Gene Expression Omnibus, GSE32317 |

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
