## [Decision Letter]

[**Editorial note:** This article has been through an editorial process in which the authors decide how to respond to the issues raised during peer review. The Reviewing Editor's assessment is that all the issues have been addressed.]

Thank you for submitting your article "IgE-mediated mast cell activation promotes inflammation and cartilage destruction in osteoarthritis" for consideration by *eLife*. Your article has been reviewed by two peer reviewers, and the evaluation has been overseen by a Reviewing Editor and Tadatsugu Taniguchi as the Senior Editor. The reviewers have opted to remain anonymous.

The Reviewing Editor has highlighted the concerns that require revision and/or responses, and we have included the separate reviews below for your consideration. If you have any questions, please do not hesitate to contact us.

Summary:

Based on the findings that abnormally degranulated mast cells exist in murine osteoarthritis (OA), in this manuscript, authors have tried to clarify how the mast cell participates in OA. They presented evidence for involvement of IgE-FceRI-Syk axis in experimental OA in mice. As seen in reviewers' comments, both of them found interest, but one reviewer requires functional assessment of the mice OA model. The other reviewer strongly requires epidemiologcal data in regard to correlation of the IgE level with higher OA risk in the human, and in vivo relationship between mast cells and various biological outputs, and between Syk in mast cells and these outputs.

Major concerns:

1) Functional assessment of the joint such as gait function or evaluating pain in mice OA model is required.

2) The relationship between IgE-derived mast cell activation and complement activation in the OA model should be, at least, properly discussed.

3) If IgE is implicated in the pathogenesis of human OA, it is reasonable to speculate the relathinship between IgE level and higher OS risk in human. Thus, these epidemiological data should be examined and provided.

4) In vivo relationship between mast cell deficiency and cytokines/proteases/fibrobalstic proliferation/solubleglycoaminoglycans in the synovial tissue is required.

5) In vivo relationship between IgE/FceRI/Syk axis and tryptase activity is required.

6) In vivo relationship between Syk in mast cells, DMM-induced cartilage destruction, the related signatures is required.

7) In regard to the data presented in this manuscript, both reviewers require more high quality, more quantification, and incorporation of suitable controls.

Separate reviews (please respond to each point):

*Reviewer #1:*

The authors present convincing evidence of the involvement of mast cells and the IgE/FceRI/Syk axis in inflammation and cartilage destruction in experimental Osteoarthritis (OA) in mice. They also described increased mast cell degranulation, mast cell tryptase release, and the upregulation of mast-cell related genes, in joints from osteoarthritis patients than in joints from non-OA trauma patients. Their findings support a role for IgE-mediated mast cell activation in OA.

Their thorough experimental approach used a variety of genetically modified mice, cell transfers, neutralizing antibodies and pharmacological treatments to assess the role of mast cells in OA. They found that the deficiency of mast cells, IgE or FceRI, and the inhibition of Syk, decreased pathology in mice OA. They also demonstrated a pathogenic activity of mast-cell tryptase in promoting inflammation and cartilage destruction in in vitro and in vivo.

What I found missing in the analysis of mice with OA is the functional assessment of the disease joint in the various experimental conditions, as they did for example in a previous publication by measuring gait function (Wang et al., 2011), or by evaluating pain (Huesa et al., 2016).

I wonder if the authors have explored the relationship between mast cell activation and complement activation in OA. Some years ago they described a central role of complement activation in human and mouse OA (Nature Medicine 2011). Since the activation of complement produces anaphylotoxins that can recruit and activate mast cells, it is possible that complement regulates mast cells in the synovia. And reciprocally, products of the degradation of cartilage by mast-cell tryptase may activate complement in the synovia.

It will be interesting to know what is temporal relationship during disease development between mast activation/degranulation and complement activation, and how deficiency or hyper-activation of one of these two mechanisms affect the other. The relationship between these two pathogenic mechanisms in human OA could also be explored, they may go together or define different disease phenotypes.

Minor Comments:

Histological analysis should be described in more detail, how many sections and microscope field were analyzed, and the criteria for grading.

*Reviewer #2:*

Osteoarthritis (OA) is a degenerative joint disease and is the most common form of arthritis. In this manuscript, the authors found abnormally degranulated mast cells in human and murine osteoarthritic joint tissue. Disruption of IgE/FceRI/Syk signaling axis ameliorated development of osteoarthritis. This signaling, along with histamine-releasing factor (HFR) promoted releases of tryptases from mast cell and induced inflammation and cartilage destruction in synovial tissues. Therefore, the authors concluded that the aberrant mast cell activation is critical for pathogenesis of OA. Despite potential importance of mast cell in OA, the present study has several concerns.

Major concerns:

In general, OA is induced by long lasting mechanical stresses and can be accompanied by chronic inflammation, whereas IgE mediates antigen-specific acute and drastic inflammatory responses in allergic diseases. Therefore, the authors need to provide very careful experimental data to show that IgE-mediated immune responses are implicated in OA pathogenesis and induction of chronic inflammation that accompanies OA. Also, antigen specificity of the IgE involved in the pathogenesis of OA is unclear. Thus, the authors have to give convincing explanation by clarifying the mechanism by which mechanical stress induces IgE production, and identifying the antigen specificity of the IgE.

1) If IgE is implicated in the pathogenesis of OA, diseases related to high level of IgE (asthma, atopic dermatitis, hay fever etc.) should be associated with the onset and severity of OA. Therefore, the authors should provide epidemiological evidence on this point.

2) A mechanical damage induced OA model does not always reflect all the features of human OA. To propose the surprising concept like IgE involvement, one model of OA is not sufficient. The authors should verify the observation as described in the manuscript by another OA model.

3) The quality of electron microscopy is too poor to distinguish the mast cell granules. The authors should provide better quality immune-electron microscopic images as shown in Figure 1.

4) The quantification of the infiltrated mast cells is only based on the unclear images (Figure 1). Likewise, mast cell quantification in destabilization of the medial meniscus (DMM)-induced mice were also determined by histological analysis in Figure 2—figure supplement 3 and is not performed in Figure 3, 4 and 5. The authors have to quantify the number of mast cells infiltrated into synovial tissues regarding patient with OA in Figure 1 and DMM-induce mice in Figure 2—figure supplement 3, Figure 3, 4 and 5 by use of FACS. In addition, morphology of the FACS-sorted mast cells from synovial tissues must be examined by appropriate staining.

5) It also remains unclear whether DMM induces accumulation of mast cell at the synovial tissues. The author should compare the number of synovial mast cells in the DMM-induced mice with those of sham operation.

6) The results shown in Figure 3 are not enough to support authors' conclusion because the experiment in which human cartilage explants are cultured in the presence of recombinant tryptase is too artificial. To prove whether the mast cell derived tryptases destroy synovial cartilage, the authors have to culture human cartilage explants with mast cells and examine the all parameters as shown in Figure 3 A-F as well as soluble glycosaminoglycans.

7) There is no evidence whether the mast cell derived tryptases are involved in the induction of proinflammatory cytokines, proteases expression, fibroblast proliferation and soluble glycosaminoglycans in vivo. The authors have to check whether mast cell deficiency and APC366 administration reduce the induction of proinflammatory cytokines, proteases expression, fibroblast proliferation and soluble glycosaminoglycans in the synovial tissues under the DMM-induced condition.

8) The authors should show the evidence that chondrocyte undergoes apoptosis.

9) There is no evidence as to whether disruption of the IgE/FceRI/Syk signaling suppresses tryptases production in vivo. The authors have to measure the tryptase activity and soluble glycosaminoglycans in the synovial fluids in Figure 4 and 5.

10) The author must examine whether DMM induces increases in IgE concentration in synovial fluids in comparison to those of sham operation. If so, is it sufficient to stimulate synovial mast cells?

11) The authors must examine whether the DMM-induced cartilage destruction as well as the related signatures (proinflammatory cytokines, proteases expression, fibroblast proliferation and soluble glycosaminoglycans) are canceled in the Cpa3-Cre Syk-flox mice.

[Editors' note: further revisions were suggested, as described below.]

Thank you for submitting the revised manuscript, entitled "IgE-mediated mast cell activation promotes inflammation and cartilage destruction in osteoarthritis". We are pleased to inform that your article is basically acceptable for publication in *eLife*, provided that you will address some minor issues.

As you can see below, one reviewer had some concerns regarding the correlation between IgE level and severeity of osteoarthritis. Regarding this concern, I judge the Ig-7KO, FceRI KO, and anti-IgE data to provide convincing genetic evidence, and as mentioned in your letter, that authors' explanation of why the IgE level did not significantly change is reasonable. Moreover, the authors carefully mentioned, for instance, "using genetic and pharmacological approaches" in the Abstract.

Please address the minor concerns of Reviewer #2 outlined below.

*Reviewer #1:*

The authors have answered a good part of the concerns of the reviewers. The manuscript is improved and I recommend that it will be accepted for publication.

*Reviewer #2:*

The authors proposed that the mast cell activation mediated by IgE/FceRI/Syk signaling is responsible for pathogenesis of OA. The authors corrected some statements in the revised manuscript. However, as I mentioned below, the fundamental problems of this manuscript have not yet been solved. In addition, they did not perform additional experiments to strengthen their conclusion.

The authors showed that bulk IgE level in serum has no correlation with severity of osteoarthritis. ELISA analysis also demonstrated that synovial fluid IgE level in patients with OA was lower than that of patients with RA and PsA. (In addition, the ELISA analysis lacks appropriate controls such as healthy control.) These results made me feel that it is very unlikely that IgE is related to development of OA. The authors also referred to the possibility that antigen specificity of IgE rather than the amount of IgE may be involved in development of OA by showing the preliminary data. However, the data are too preliminary to convince us of the surprising finding that OA is an autoimmune disease caused by IgE against join antigen(s) and mast cells. The proposed mechanism is not well based on evidence. Taken together, I cannot recommend this manuscript for the publication in *eLife*.

1) In general, OA is induced by long lasting mechanical stresses and can be accompanied by chronic inflammation, whereas IgE mediates antigen-specific acute and drastic inflammatory responses in allergic diseases. Thus, it is not easy to imagine that IgE-mediated immune responses are implicated in OA pathogenesis and induction of chronic inflammation that accompanies OA. The authors have to give convincing explanation.

2) Authors showed that IgE level does not corelate with OA onset or severity. I think that this is contradictory to authors' conclusion. Authors tried to explain this by the antigen-specific IgE, but it is unbelievable that OA is an autoimmune disease caused by IgE against join antigen(s). If so, they should identify the specific antigens recognized by IgE from the synovial tissues.

3) The authors could not analyze synovial derived mast cells by flow cytometry because enzymatic dissociation of synovium physically destroys mast cells. But many researchers succeeded in cell preparation from human and mouse synovial tissue. In particular, FACS analysis of mast cells in synovial tissues was previously shown (Lee et al., Arthritis Rheum., 2013). Therefore, the authors must show the accurate number of mast cell by FACS in Figure 1 to 5 and Figure 2—figure supplement 1.

4) The authors cannot conclude that mast cell-derived tryptases promote osteoarthritis-associated pathology, since it remained unclear whether tryptase secreted by mast cells is sufficient to induce the destruction of the synovial cartilage. As I pointed out in the comment #6, addition of huge amount of tryptase is too artificial. The authors have to perform the co-culture experiment of mast cells with human cartilage explants. Furthermore, the authors have to show the evidence that chondrocyte undergoes apoptosis as I mentioned in comment #8.

---

## [Author Response]

Major concerns:1) Functional assessment of the joint such as gait function or evaluating pain in mice OA model is required.

The primary goal of our study was to demonstrate that mast cells play a crucial role in the pathogenesis of osteoarthritis, and specifically in the mechanisms that result in cartilage and joint destruction. Although it was previously reported that mast cells contribute to osteoarthritis joint pain in mice (Sousa-Valente et al., 2018) and that synovial mast cell numbers are associated with the degree of synovitis in osteoarthritic joints in humans (de Lange-Brokaar et al., 2016), mast cells have not previously been demonstrated to directly contribute to cartilage and joint tissue breakdown in osteoarthritis. In this manuscript, we demonstrate that IgE-mediated mast cell activation plays a critical role in cartilage degeneration, joint destruction and the development of osteoarthritis.

We and others previously established that the cartilage degeneration and joint destruction that develop following DMM in mice are associated with abnormal gait and increased pain (Huesa et al., 2016; Wang et al., 2011). Although we recognize the importance of pain and functional outcomes, we believe that histologic assessment of cartilage degeneration and joint destruction are the most definitive outcomes in mouse DMM osteoarthritis studies. Due to the large number of different murine experiments performed as part of the studies described in this manuscript, our analyses were focused on the definitive histologic outcome measures and we did not perform pain and function assessments. We now address this limitation in the Discussion:

“There are multiple potential limitations to this study. First, it was previously shown that mast cells are associated with joint pain in murine osteoarthritis (Sousa-Valente et al., 2018) and that synovial mast cell numbers are associated with the degree of synovitis in human osteoarthritis (de Lange-Brokaar et al., 2016). We and others previously demonstrated that following DMM in mice development of histologic osteoarthritis is associated with poor functional outcomes including pain and abnormal gait (Huesa et al., 2016; Wang et al., 2011). The studies presented in this manuscript demonstrate that mast cells and dysregulated mast cell activation contribute to cartilage and joint degeneration following DMM, and based on this prior work (Huesa et al., 2016; Wang et al., 2011) such pathologic changes are anticipated to result in pain and abnormal gait. Future studies will be needed to further characterize the role of mast cells in osteoarthritis-associated pain and joint dysfunction…”

2) The relationship between IgE-derived mast cell activation and complement activation in the OA model should be, at least, properly discussed.

We agree with the reviewers that the connection between mast cell activation by IgE and complement should be discussed. We previously demonstrated that complement components are dysregulated in osteoarthritis and contribute to osteoarthritis pathogenesis, and it is possible that activated complement components stimulate mast cell recruitment and/or activation. We amended the Discussion section to address these possibilities:

“We previously demonstrated that complement plays a critical role in the pathogenesis of osteoarthritis (Lepus et al., 2014; Wang et al., 2011). Activation of the complement system results in the production of C3a and C5a which serve as anaphylatoxins and activators of mast cells (Erdei et al., 2004; Gaudenzio et al., 2016). It is therefore possible that complement regulates the recruitment and activation of mast cells in the synovium to promote the pathogenesis of osteoarthritis. We previously demonstrated that the cartilage breakdown product, fibromodulin, can activate the complement system (Wang et al., 2011), and it is possible that mast cell tryptase-mediated cartilage degeneration produces fibromodulin and other cartilage breakdown products that reciprocally activate complement in the synovia in osteoarthritis. It will be important to define the temporal relationship between mast cell activation/degranulation and complement activation in the development and progression of osteoarthritis, including how activation of one affects the other and vice versa. Further, complement is capable of stimulating mast cells alone, and can also enhance IgE-dependent mast cell activation and degranulation (Schafer et al., 2013). Future studies will be needed to further define the roles and relationships of complement activation, IgE, and mast cell activation/degranulation in the development and progression of osteoarthritis.”

3) If IgE is implicated in the pathogenesis of human OA, it is reasonable to speculate the relathinship between IgE level and higher OS risk in human. Thus, these epidemiological data should be examined and provided.

We appreciate the reviewers’ comment. However, previous data from others, as well as new data that we present in this response, suggest that levels of total IgE in either human sera or synovial fluids are not associated with osteoarthritis risk or severity. Previous reports found that IgE levels are not elevated in the synovial fluids of osteoarthritis joints (Hunder and Gleich, 1974). In addition, while increased serum levels of IgE can be detected in some atopic diseases, this is also not always the case (Novak and Bieber, 2003). Further, in allergic diseases it is antigen-specific IgE, and not bulk IgE, that plays a critical role in activating mast cells. Thus, it is possible that antigen-specific IgE, rather than bulk IgE, might play a role in the pathogenesis of osteoarthritis.

To identify potential correlations of bulk IgE with osteoarthritis in humans, we analyzed the relationship between serum IgE levels and the presence and severity of osteoarthritis. In collaboration with David Felson and using serum samples from 1,200+ individuals in the Framingham Osteoarthritis Study (Felson et al., 1987), we measured levels of serum IgE and performed statistical analyses to determine whether bulk serum IgE was associated with the presence of osteoarthritis and/or osteoarthritis severity. At certain serum IgE thresholds (e.g. >200 ng/mL), we detected trends towards more severe osteoarthritis in individuals with increased serum IgE, but these results were not statistically significant (Author response image 1; p=0.08). We were not able to demonstrate a statistical association of bulk IgE levels in serum with the severity of osteoarthritis (based on KL score; data not shown). Further analyses using even larger patient numbers may enable demonstration of a modest association of serum IgE levels with osteoarthritis and/or osteoarthritis severity.

We also used ELISA to measure bulk IgE levels in osteoarthritis (OA), rheumatoid arthritis (RA), and psoriatic arthritis (PsA) synovial fluids (Author response image 2). In these analyses, we also did not observe evidence of increased IgE levels in osteoarthritis synovial fluids as compared to the other subgroups.

**Author response image 2. respfig2:** 

It remains possible that antigen-specific IgE, rather than bulk IgE levels, will be associated with the development of osteoarthritis. To address this possibility, we examined whether osteoarthritis synovial fluid IgE is able to bind cartilage extracts. We found that osteoarthritic joint synovial fluid IgE can bind both damaged “lesional” cartilage and “non-lesional” cartilage from osteoarthritic joints (Author response image 3).

**Author response image 3. respfig3:** 

These data suggest that antigen-specific IgE might play a role in the pathogenesis of osteoarthritis. Nevertheless, we believe these findings are too preliminary to include in the main manuscript. Further work will be needed to validate these preliminary findings, to define the role of antigen-specific IgE, and to characterize the antigen targets of IgE, in osteoarthritis.

We added the following text to the Discussion to address our preliminary findings and these possibilities:

“Previous reports found that IgE levels are not elevated in the synovial fluids of osteoarthritis joints (Hunder and Gleich, 1974), and in preliminary studies we observed only trends towards an association of increased bulk IgE in serum with osteoarthritis in humans. It remains possible that antigen-specific IgE contributes to mast cell activation in osteoarthritis, and future studies will be needed to investigate the relative contributions of antigen-specific IgE and other mechanisms of mast cell activation in the pathogenesis of osteoarthritis.”

*4)* In vivo *relationship between mast cell deficiency and cytokines/proteases/fibrobalstic proliferation/solubleglycoaminoglycans in the synovial tissue is required.*

*5)* In vivo *relationship between IgE/FceRI/Syk axis and tryptase activity is required.*

*6)* In vivo *relationship between Syk in mast cells, DMM-induced cartilage destruction, the related signatures is required.*

We appreciate the reviewers’ suggestions in these three interrelated questions. To provide further mechanistic data regarding the role of Syk in mast cell activation, we performed new in vivo experiments to examine the expression of a panel of mast cell mediators during treatment with the Syk inhibitor PRT062607 following DMM. We treated mice with either PRT062607 or vehicle for 6-weeks following DMM, and then harvested DMM joint synovial tissue and used qPCR to analyze expression levels of mRNAs encoding pro-inflammatory cytokines and proteases. We found that expression of multiple pro-inflammatory cytokines, chemokines and proteases were reduced in DMM joint tissues derived from mice receiving pharmacologic treatment with the Syk inhibitor as compared to vehicle. We now include these data in new Figure 5I. The following text was added to the “IgE signaling through FcRI promotes pathogenesis of osteoarthritis” sub-section of the Results:

“We used qPCR to analyze the levels of mRNAs encoding pro-inflammatory cytokines and degradative enzymes known to be produced by mast cells. Six-weeks following DMM, transcriptional expression of multiple pro-inflammatory cytokines and proteases including IL-1, CCL2, ADAMTS4 and MMP13 were significantly reduced in DMM joint tissues derived from mice treated with the Syk-inhibitor PRT062607 as compared to vehicle (Figure 5I).”

Together, the data in our manuscript demonstrate a critical role for mast cells and mast cell-derived tryptases in promoting cartilage degeneration and joint destruction following DMM in mice. We specifically demonstrate in DMM mice that mast cell-deficiency (Figure 2A-D), treatment with inhibitors of mast cell survival or signaling (Figure 2E-H; Figure 5E-H), deficiency or depletion of IgE (Figure 4), and treatment with the APC366 inhibitor of mast cell tryptase (Figure 3G-J), all reduce cartilage destruction (Figures 2B, 2F, 3H, 4B, 4F, 5F) and synovitis (Figures 2D, 2H, 3J, 4D, 4H, 5D). We appreciate the reviewer’s suggestion for inclusion of additional data linking mast cells and mast cell activation pathways to the production of tryptase and other mast cell and inflammatory mediators, but believe our data provide compelling evidence for a critical role for mast cells and mast cell tryptases in the pathogenesis of osteoarthritis. Further, the DMM model requires a protracted timeline – mice must be aged to 18-20 weeks prior to surgical DMM, followed by a 4 month in-life period following surgical DMM prior to harvesting of joint tissues, and then 1+ month is necessary for histologic processing of joint tissues – making it not logistically possible for us to perform the additional experiments necessary to address all of the reviewers’ suggestion in the time allotted. As a result, we now address the need for further investigation of the role of mast cells in promoting these changes as a limitation of our study in the Discussion:

“There are multiple potential limitations to our study… Second, while our findings demonstrate a critical role for mast cells, mast cell activation pathways, and the mast cell product tryptase in the development of osteoarthritis following DMM, further investigation is needed to characterize the relationship between IgE-mediated mast cell activation and the presence of pro-inflammatory cytokines and proteases implicated in synovitis and cartilage degradation in osteoarthritis.”

7) In regard to the data presented in this manuscript, both reviewers require more high quality, more quantification, and incorporation of suitable controls.

We revised the manuscript to address these concerns from the reviewers. The findings from our new experiments further strengthen our conclusions that IgE-mediated mast cell activation promotes the pathogenesis of osteoarthritis. The following changes and/or new data were added:

1) We replaced the previous images from our electron microscopy analysis of mast cell degranulation and tryptase release in synovial tissues from humans with osteoarthritis in Figure 1C with new higher-quality versions of these images.

2) In response to comment #7 from reviewer #2, we now provide additional mechanistic data implicating mast cell-derived tryptases in the pathogenesis of osteoarthritis by demonstrating that in DMM synovial tissue mRNA levels of multiple pro-inflammatory cytokines and degradative enzymes are reduced by pharmacologic inhibition of tryptase (new Figure 3E). We also now demonstrate that treatment of synovial fibroblasts with tryptase promotes pro-inflammatory signaling through activation of the MAPK/ERK pathway (new Figures 3J and K).

3) In response to Concern #6 from the Reviewing Editor and comment #11 from reviewer 2, we now provide mechanistic data supporting a key role for Syk signaling in mast cells in the pathogenesis of osteoarthritis by showing that pharmacologic inhibition of Syk reduces mRNA levels of multiple pro-inflammatory cytokines and degradative enzymes in DMM synovial tissues (new Figure 5I).

4) In response to comment #4 from reviewer #2, we now present data quantifying the numbers of mast cells present in mouse joint synovia from multiple experiments presented in the manuscript. Specifically, we quantified the numbers toluidine blue-positive mast cells in the synovia of DMM joints from two strains of mast cell-deficient mice (*Kit*^W-sh/W-sh^ and *Cpa3-Cre;MCl^-^1^fl/fl^*), including their levels following mast cell reconstitution (Figure 2—figure supplement 3B and D). We also quantified the numbers of toluidine blue-positive mast cells in synovial tissues of IgE-sufficient (*Igh7^+/+^*) and IgE-deficient (*Igh7^-/-^*) mice following DMM (Figure 4—figure supplement 2).

5) We performed new immunostains for mast cells in synovial samples from human osteoarthritis and post-trauma non-osteoarthritis joints, and demonstrate that similar numbers of mast cells are present in both conditions (Figure 1—figure supplement 2). Although there are similar numbers of mast cells in human osteoarthritis and post-trauma non-osteoarthritis synovial tissue samples, we demonstrate that the mast cells in the osteoarthritis synovium are actively degranulating (Figure 1B and C) and releasing activated tryptase into the synovial fluid (Figure 1A).

6) We now include additional controls demonstrating that the observed downregulation of these cytokines and proteases is specific and not a general response (new Figure panels 3E and 5I).

Separate reviews (please respond to each point):

Reviewer #1:

The authors present convincing evidence of the involvement of mast cells and the IgE/FceRI/Syk axis in inflammation and cartilage destruction in experimental Osteoarthritis (OA) in mice. They also described increased mast cell degranulation, mast cell tryptase release, and the upregulation of mast-cell related genes, in joints from osteoarthritis patients than in joints from non-OA trauma patients. Their findings support a role for IgE-mediated mast cell activation in OA.

*Their thorough experimental approach used a variety of genetically modified mice, cell transfers, neutralizing antibodies and pharmacological treatments to assess the role of mast cells in OA. They found that the deficiency of mast cells, IgE or FceRI, and the inhibition of Syk, decreased pathology in mice OA. They also demonstrated a pathogenic activity of mast-cell tryptase in promoting inflammation and cartilage destruction in* in vitro *and* in vivo.

What I found missing in the analysis of mice with OA is the functional assessment of the disease joint in the various experimental conditions, as they did for example in a previous publication by measuring gait function (Wang et al., 2011), or by evaluating pain (Huesa et al., 2016).

We appreciate the reviewer’s suggestion. Nevertheless, the primary objective of this study was to identify molecular mechanisms underlying structural changes in osteoarthritis. As previously demonstrated in both Wang et al., (2011) and Huesa et al., (2016), the cartilage degeneration and joint breakdown that occurs following DMM results in both abnormal gait and increased pain. We believe that histologic demonstration of cartilage degeneration, joint breakdown, and synovitis is the most definitive outcome in the DMM model, and as a result this was the focus of our studies. Additionally, a prior study implicated mast cells in human osteoarthritis pain (Sousa-Valente et al., 2018), but did not demonstrate that they play a direct role in the pathogenesis of osteoarthritis. In this manuscript, we provide extensive evidence based on histologic outcomes that mast cells and IgE play critical roles in the cartilage breakdown, joint tissue destruction, and synovitis that are characteristic of osteoarthritis. Given that we and others previously demonstrated that histologic osteoarthritis pathology is associated with pain and gait dysfunction in mice (Huesa et al., 2016; Wang et al., 2011), we do not believe that assessment of gait function or pain would significantly advance the conclusions of our study. We now address this potential limitation of our study in the Discussion:

“There are multiple potential limitations to this study. First, it was previously shown that mast cells are associated with joint pain in murine osteoarthritis (Sousa-Valente et al., 2018) and that synovial mast cell numbers are associated with the degree of synovitis in human osteoarthritis (de Lange-Brokaar et al., 2016). We and others previously demonstrated that following DMM in mice development of histologic osteoarthritis is associated with poor functional outcomes including pain and abnormal gait (Huesa et al., 2016; Wang et al., 2011). The studies presented in this manuscript demonstrate that mast cells and dysregulated mast cell activation contribute to cartilage and joint degeneration following DMM, and based on this prior work (Huesa et al., 2016; Wang et al., 2011) such pathologic changes are anticipated to result in pain and abnormal gait. Future studies will be needed to further characterize the role of mast cells in osteoarthritis-associated pain and joint dysfunction…”

I wonder if the authors have explored the relationship between mast cell activation and complement activation in OA. Some years ago they described a central role of complement activation in human and mouse OA (Nature Medicine 2011). Since the activation of complement produces anaphylotoxins that can recruit and activate mast cells, it is possible that complement regulates mast cells in the synovia. And reciprocally, products of the degradation of cartilage by mast-cell tryptase may activate complement in the synovia.It will be interesting to know what is temporal relationship during disease development between mast activation/degranulation and complement activation, and how deficiency or hyper-activation of one of these two mechanisms affect the other. The relationship between these two pathogenic mechanisms in human OA could also be explored, they may go together or define different disease phenotypes.

We appreciate the reviewer’s comments in these two related questions. We agree that the known connection between complement and mast cells makes it possible that complement regulates mast cell recruitment and activation in osteoarthritis synovia. We also agree that understanding the temporal relationship between complement and mast cell activation in the development and progression of osteoarthritis is an important question. We now address these possibilities in the following new text added to the Discussion:

“We previously demonstrated that complement plays a critical role in the pathogenesis of osteoarthritis (Lepus et al., 2014; Wang et al., 2011). Activation of the complement system results in the production of C3a and C5a which serve as anaphylatoxins and activators of mast cells (Erdei et al., 2004; Gaudenzio et al., 2016). It is therefore possible that complement regulates the recruitment and activation of mast cells in the synovium to promote the pathogenesis of osteoarthritis. We previously demonstrated that the cartilage breakdown product, fibromodulin, can activate the complement system (Wang et al., 2011), and it is possible that mast cell tryptase-mediated cartilage degeneration produces fibromodulin and other cartilage breakdown products that reciprocally activate complement in the synovia in osteoarthritis. It will be important to define the temporal relationship between mast cell activation/degranulation and complement activation in the development and progression of osteoarthritis, including how activation of one affects the other and vice versa. Further, complement is capable of stimulating mast cells alone, and can also enhance IgE-dependent mast cell activation and degranulation (Schafer et al., 2013). Future studies will be needed to further define the roles and relationships of complement activation, IgE, and mast cell activation/degranulation in the development and progression of osteoarthritis.”

Minor Comments:Histological analysis should be described in more detail, how many sections and microscope field were analyzed, and the criteria for grading.

We now include a detailed description of the histological analysis and methods of scoring cartilage degradation, osteophyte formation and synovitis. The following text was added to the “Histologic assessment of osteoarthritic development in mice” sub-section of the Materials and methods:

“Stifle joints were harvested 12 or 20 weeks after DMM and fixed in 10% neutral buffered formalin followed by decalcification in formic acid for 48 hours. Joints were then embedded in paraffin, and 6-μm sections cut from 3 separate levels of the joint and stained with Safranin-O for assessment of cartilage damage; H&E for assessment of synovial thickening (synovitis) and osteophyte formation; and toluidine blue for the assessment of mast cells. Cartilage degeneration, synovitis, and osteophyte formation were evaluated by two blinded observers using a modified version of a described scoring system (Kamekura et al., 2005) as we previously described (Wang et al., 2011). In brief: Cartilage degeneration was calculated by depth of cartilage degeneration (score of 0–4) × width of cartilage degeneration (with a score of 1 meaning one-third of the surface area, a score of 2 meaning two-thirds of the surface area, and a score of 3 meaning the whole surface area) in each third of the femoral-medial and tibial-medial condyles. The scores for the six regions were then summed. Synovitis scores were calculated as previously described (Blom et al., 2004): 0, no changes compared to normal joints; 1, thickening of the synovial lining and some influx of inflammatory cells; 2, thickening of the synovial lining and intermediate influx of inflammatory cells; and 3, profound thickening of the synovial lining (more than four cell layers) and maximal observed influx of inflammatory cells. Scores for synovitis were recorded for the femoral-medial and the tibial-medial condyles, and the scores for the two regions summed. Osteophyte formation was scored according to a previously described scoring system (Kamekura et al., 2005): 0, none; 1, formation of cartilage-like tissues; 2, increase of cartilaginous matrix; 3, endochondral ossification. Mast cells were quantified by a blinded examiner who determined the number of toluidine blue-positive mast cells per high power field of the joint sections.”

Reviewer #2:

Osteoarthritis (OA) is a degenerative joint disease and is the most common form of arthritis. In this manuscript, the authors found abnormally degranulated mast cells in human and murine osteoarthritic joint tissue. Disruption of IgE/FceRI/Syk signaling axis ameliorated development of osteoarthritis. This signaling, along with histamine-releasing factor (HFR) promoted releases of tryptases from mast cell and induced inflammation and cartilage destruction in synovial tissues. Therefore, the authors concluded that the aberrant mast cell activation is critical for pathogenesis of OA. Despite potential importance of mast cell in OA, the present study has several concerns.Major concerns:In general, OA is induced by long lasting mechanical stresses and can be accompanied by chronic inflammation, whereas IgE mediates antigen-specific acute and drastic inflammatory responses in allergic diseases. Therefore, the authors need to provide very careful experimental data to show that IgE-mediated immune responses are implicated in OA pathogenesis and induction of chronic inflammation that accompanies OA. Also, antigen specificity of the IgE involved in the pathogenesis of OA is unclear. Thus, the authors have to give convincing explanation by clarifying the mechanism by which mechanical stress induces IgE production, and identifying the antigen specificity of the IgE.1) If IgE is implicated in the pathogenesis of OA, diseases related to high level of IgE (asthma, atopic dermatitis, hay fever etc.) should be associated with the onset and severity of OA. Therefore, the authors should provide epidemiological evidence on this point.

We appreciate the reviewer’s suggestion. The possibility of osteoarthritis being associated with allergic diseases is an interesting question. While an epidemiological analysis could provide important new insights, we believe an epidemiological analysis is beyond the scope of our current study. We now address these possibilities in the following text added to the Discussion:

“While many classical IgE-mediated allergic diseases including asthma, allergic rhinitis and eczema exhibit comorbidities (Pedersen and Weeke, 1983), we are not aware of evidence of a clear link between osteoarthritis and allergic diseases. There is significant evidence that these classic allergic diseases are caused by antigen-specific IgE-dependent activation of mast cells (Galli and Tsai, 2012; Hamelmann et al., 1997; Oettgen and Geha, 2001; van der Heijden et al., 1993). If the role of mast cells in osteoarthritis pathogenesis is dependent on antigen-specific IgE, the target antigens could potentially be exogenous allergens. Another possibility is that in osteoarthritis the IgE target antigens are bone or cartilage breakdown products that give rise to neoantigens following joint injury or instability. Although we are not aware of evidence that osteoarthritis is associated with classic allergic and/or IgE-dependent diseases, an epidemiological analysis that addresses this important question is warranted. Further studies are also needed to characterize the antigen targets of IgE in osteoarthritis, which could reveal shared or distinct targets as compared to classical allergic diseases as well as provide insights into the mechanistic basis for the development of osteoarthritis.”

2) A mechanical damage induced OA model does not always reflect all the features of human OA. To propose the surprising concept like IgE involvement, one model of OA is not sufficient. The authors should verify the observation as described in the manuscript by another OA model.

We appreciate the reviewer’s comment. While no mouse model perfectly mimics osteoarthritis pathogenesis in humans, we chose to use the DMM model because it is highly representative of a significant subset of human osteoarthritis and is also the gold standard in the field for experimental investigation of osteoarthritis (Culley et al., 2015; Lorenz and Grässel, 2014). We, however, note that the DMM model is most representative of osteoarthritis development following traumatic joint injury (PTOA) in humans. It is possible that non-traumatic osteoarthritis, calcium pyrophosphate crystal-associated osteoarthritis, or other subsets of osteoarthritis may arise from activation of other molecular pathways. Thus, IgE-dependent mast cell activation may not contribute to development of all subsets of osteoarthritis disease.

We previously demonstrated that complement activation promotes osteoarthritis pathogenesis in both the medial meniscectomy (MM) and destabilization of the medial meniscus (DMM) mechanical injury mouse models (Wang et al., 2011), indicating that similar molecular mechanisms related to the complement system underlie osteoarthritis pathogenesis in both models. To test whether IgE activation similarly promotes osteoarthritis in multiple mouse models, we have now analyzed the effect of genetic deficiency of the high-affinity IgE receptor, Fcer1 (*Fcer1^-/-^*) on the development of osteoarthritis in the MM mouse model. Following MM, *Fcer1^-/-^* mice were protected from cartilage degeneration as compared to wild-type mice (Author response image 4). This data recapitulates our results in the DMM model (Figures 4 and 5), indicating that IgE-signaling promotes osteoarthritis pathogenesis in multiple mechanical injury models.

We now clarify that based on our use of the DMM model, our data suggest a key role for IgE in the pathogenesis of PTOA. The following text was added to the Discussion:

“Together, our results demonstrate that the IgE/FceR1/Syk axis-activated mast cells to promote the development of osteoarthritis following mechanical injury (DMM) in mice. As the DMM model is most representative of osteoarthritis development following traumatic joint injury (PTOA) in humans, these findings suggest that PTOA arises as result of activation of the IgE/FceRI/Syk axis. It is possible that non-traumatic osteoarthritis, calcium pyrophosphate crystal-associated osteoarthritis, or other subsets of osteoarthritis may arise from activation of other molecular pathways.”

**Author response image 4. respfig4:** Fcer1-deficient mice are protected against the medial meniscectomy (MM) model of osteoarthritis. Representative toluidine blue stained medial stifle joint sections from Fcer1-deficient (*Fcer1^-/-^*) and wild-type (*Fcer1^+/+^*) mice; arrowheads show areas severe cartilage loss (upper panels). Quantification of cartilage degeneration in the medial regions of stifle joints from these mice (lower panel).

3) The quality of electron microscopy is too poor to distinguish the mast cell granules. The authors should provide better quality immune-electron microscopic images as shown in Figure 1.

We now included higher-quality versions of images from our election microscopy analysis of tryptase in synovial tissues from osteoarthritis and healthy individuals in Figure 1B.

4) The quantification of the infiltrated mast cells is only based on the unclear images (Figure 1). Likewise, mast cell quantification in destabilization of the medial meniscus (DMM)-induced mice were also determined by histological analysis in Figure 2—figure supplement 3 and is not performed in Figure 3, 4 and 5. The authors have to quantify the number of mast cells infiltrated into synovial tissues regarding patient with OA in Figure 1 and DMM-induce mice in Figure 2—figure supplement 3, Figure 3, 4 and 5 by use of FACS. In addition, morphology of the FACS-sorted mast cells from synovial tissues must be examined by appropriate staining.

The images provided in Figure 1B are representative from the overall experiment presented as well as from 2 additional independent experiments that utilized independent osteoarthritis and non-osteoarthritis samples. In the legend for Figure 1, we indicate that for the representative experiment presented in the manuscript, mast cells and degranulation were quantified in samples from 5 individuals with osteoarthritis and 5 non-osteoarthritis individuals. Quantitation of the blinded scoring of mast cell degranulation in human osteoarthritis vs. post-traumatic non-osteoarthritis synovium is presented in Figure 1C (P <0.01). Further, it is important to note that we also demonstrated increased levels of activated tryptase in human osteoarthritis as compared to non-osteoarthritis synovial fluids (Figure 1A), showing active and ongoing degranulation of mast cells in human osteoarthritis synovium, given the half-life of activated tryptase is approximately 2 hours.

In regard to the quantification of mast cell numbers in humans, others reported that synovial mast cell numbers are (i) elevated in osteoarthritis as compared to healthy synovium (Dean et al., 1993), and (ii) associated with the degree of synovitis in human osteoarthritis (de Lange-Brokaar et al., 2016).

Performance of FACS on synovial cells requires enzymatic digestion and dissociation of the synovium into single cell suspensions. We, and the NIH Accelerating Medicines Partnerships consortium of which we are a participating member, have found that such enzymatic dissociation of synovium (with Liberase TL and/or other enzymes, either with or without the gentleMACS Dissociator) physically destroys mast cells, thereby limiting their recovery. Such makes it technically difficult and likely inaccurate to quantitate synovial mast cells through FACS, and we believe electron microscopy and immunostaining represent more appropriate methods. To further assess mast cell levels in human osteoarthritis synovial tissues, we performed new immunostains for mast cells on synovium from joints with osteoarthritis (OA) and from joints sustaining traumatic joint injury but exhibiting no radiographic evidence of osteoarthritis (post-trauma, non-OA). These immunostains demonstrate similar levels of mast cells in the synovium from both groups (New Figure 1—figure supplement 2). We have not compared the levels of mast cells observed in osteoarthritis with normal synovium, nor with the degree of synovitis in osteoarthritis synovium, as reported by others (de Lange-Brokaar et al., 2016; Dean et al., 1993). We note that our studies reported in this manuscript are focused on mast cell activation and degranulation in osteoarthritis, and we provide evidence of ongoing mast cell activation and degranulation in human osteoarthritis through both electron microscopy demonstration of actively degranulating mast cells in osteoarthritis synovium and ELISA demonstration of increased activated mast cell tryptase (a mast cell degranulation product) in osteoarthritis synovial fluids (Figure 1A-C; Figure 1—figure supplement 3).

We now also provide new data quantifying the numbers of mast cells present in synovial tissues derived from our DMM experiments in Figure 2 and Figure 2—figure supplement 1 for which images are presented in Figure 2—figure supplement 3. For this quantitation, a blinded examiner determined the number of toluidine blue-positive mast cells per high power field in synovium derived from each of the mice included in the experiments presented in Figure 2 (Figure 2—figure supplement 3). This analysis demonstrates statistically decreased mast cells in both the *Kit*^W-sh/W-sh^ and *Cpa3-Cre;MCl^-^1^fl/fl^*mast cell-deficient mice as compared to both wild-type mice and the mast cell engrafted mice (P < 0.01; new Figure 2—figure supplement 3 B and D). Toluidine blue staining of mast cells in DMM synovium was confirmed by anti-tryptase immunostaining (Figure 2—figure supplement 3E). We also demonstrate that imatinib-mediated protection against osteoarthritis following DMM was associated with reduced mast cell numbers in mouse DMM synovium (Figure 2—figure supplement 4 B – C). Further, we now provide new data quantifying the numbers of mast cells in the IgE-sufficient (*Igh7^+/+^*) and IgE-deficient (*Igh7^-/-^*) mice from our DMM experiments in Figure 4A-D. These data demonstrate decreased numbers of mast cells in *Igh7^-/-^* as compared to *Igh7^+/+^* mice (P < 0.05; new Figure 4—figure supplement 2).

5) It also remains unclear whether DMM induces accumulation of mast cell at the synovial tissues. The author should compare the number of synovial mast cells in the DMM-induced mice with those of sham operation.

We agree with the reviewer that this is an important experiment. We now quantify the numbers of mast cells present in synovial tissues of wild-type mice 20-weeks following either sham or DMM surgery. For this quantitation, a blinded examiner determined the number of toluidine blue-positive mast cells per high power section for each of the mice in this experiment. These data demonstrate a trend toward increased numbers of mast cells in DMM synovial tissues as compared to Sham synovial tissues, however this difference is not statistically significant (Author response image 5).

**Author response image 5. respfig5:** Quantitation of mast cells by toluidine blue staining of joint tissue derived from sham- and DMM-operated wild-type mice 20-weeks following surgical DMM. Statistical comparisons were performed using Student’s t test (N.S.= not significant).

6) The results shown in Figure 3 are not enough to support authors' conclusion because the experiment in which human cartilage explants are cultured in the presence of recombinant tryptase is too artificial. To prove whether the mast cell derived tryptases destroy synovial cartilage, the authors have to culture human cartilage explants with mast cells and examine the all parameters as shown in Figure 3 A-F as well as soluble glycosaminoglycans.

We agree with the reviewer that the in vitro human cartilage explant experiments represent a highly artificial experimental system, and as a result have now removed these experiments and the resulting data from the manuscript (old Figures 3A and B). Figures 3G and H, in which treatment of DMM mice with the tryptase inhibitor, APC366, preserves cartilage in vivo, show that tryptases promote cartilage degradation in osteoarthritis. We have now moved the mouse DMM experiment with treatment of APC366 (originally g-j) to Figures 3A-D. We also added new experimental data demonstrating that pharmacologic inhibition of tryptase results in decreased transcriptional expression of inflammatory and degradative mast cell mediators in DMM joint tissues (new Figure 3E; see response to the next comment).

*7) There is no evidence whether the mast cell derived tryptases are involved in the induction of proinflammatory cytokines, proteases expression, fibroblast proliferation and soluble glycosaminoglycans* in vivo*. The authors have to check whether mast cell deficiency and APC366 administration reduce the induction of proinflammatory cytokines, proteases expression, fibroblast proliferation and soluble glycosaminoglycans in the synovial tissues under the DMM-induced condition.*

We agree with the reviewer that analysis of pharmacologic inhibition of tryptase on the expression of pro-inflammatory mast cell mediators would further strengthen our findings. To address this, we performed new experiments measuring the mRNA levels in DMM joint tissues of pro-inflammatory cytokines, chemokines and proteases following treatment with the tryptase inhibitor APC366. Six-week following DMM, transcriptional expression of multiple pro-inflammatory cytokines, chemokines and proteases was reduced in DMM joint tissues derived from mice treated with the tryptase inhibitor APC366 as compared to vehicle. We now include this analysis as new Figure 3E.

We further investigated the role of tryptase in promoting pro-inflammatory signaling in synovial fibroblasts by treating cultured primary synovial fibroblasts from individuals with osteoarthritis with purified tryptase. Western blot analysis shows that phosphorylation of Erk1/2 was stimulated following treatment with tryptase. We now include this analysis as new Figures 3J and K.

Based on the changes to Figure 3, we revised the “Mast cell-derived tryptases promote osteoarthritis-associated pathology” sub-section of the Results:

“Having established a pathogenic role for mast cells in osteoarthritis and because levels of the activated form of mast cell-derived tryptase are significantly elevated in the synovial fluids of individuals with osteoarthritis, a finding in agreement with previous reports (Nakano et al., 2007), we next investigated mechanisms by which tryptase might promote the pathogenesis of osteoarthritis. We first tested whether selectively inhibiting the protease activity of tryptase with APC366 (Cairns, 2005) – an oral, selective tryptase small-molecule inhibitor previously used to alleviate allergic, inflammatory and fibrotic responses in multiple mouse models (Lu et al., 2014; Matos et al., 2013; Sevigny et al., 2011) – could effectively attenuate the progression and/or severity of osteoarthritis in mice. We found that following DMM, treatment with APC366 for 12 weeks significantly reduced cartilage damage (Figure 3A and B), osteophyte formation (Figure 3C, Figure 3 —figure supplement 2) and synovitis (Figure 3D, Figure 3 —figure supplement 2) compared to treatment with the vehicle-control mice, suggesting that tryptase inhibition can prevent the development of osteoarthritis in mice. We, additionally, measured the expression of pro-inflammatory and degradative mediators known to be produced by mast cells in DMM joints following treatment with the tryptase inhibitor APC366. Six-weeks after DMM, transcriptional expression of multiple mediators including IL-1, IL-6, IL-8, CCL2, CCL5, ADAMTS4 and *MMP3* was significantly reduced in DMM synovial tissues derived from tryptase treated as compared to vehicle treated mice (Figure 3E).

As tryptase has been shown to promote pathogenic properties in human rheumatoid arthritis-derived synovial fibroblasts (Xue et al., 2012), we examined whether tryptase could also induce pro-inflammatory and proliferative responses in primary synovial fibroblasts derived from remnant osteoarthritic joint tissue. Indeed, tryptase significantly increased the expression of the pro-inflammatory cytokine IL-1 and degradative enzymes *MMP3* and ADAMTS4 (Figure 3F), increased the secretion of cytokines IL-1β (Figure 3G), IFN (Figure 3H), and increased synovial fibroblast proliferation in vitro, as demonstrated by increased expression of the activation marker Ki-67 by fibroblasts (Figure 3I). in vitro treatment of synovial fibroblasts with tryptase also promoted phosphorylation of Erk1/2, indicating that tryptase can activate pro-inflammatory signaling pathways in synovial fibroblasts (Figure 3J and K). Further, in vitro inhibition of tryptase activity with APC366 abrogated the pro-inflammatory and proliferative responses of synovial fibroblasts (Figure 3F-I).”

8) The authors should show the evidence that chondrocyte undergoes apoptosis.

Based on this reviewer’s comment #6, we have now removed the data from our in vitro experiments treating cultured chondrocytes with tryptase (original Figures 3A and B). As a result, this suggestion is no longer relevant.

*9) There is no evidence as to whether disruption of the IgE/FceRI/Syk signaling suppresses tryptases production* in vivo*. The authors have to measure the tryptase activity and soluble glycosaminoglycans in the synovial fluids in Figure 4 and 5.*

We appreciate the reviewer’s comment and believe this analysis is important. However, the mouse experiments required to analyze the relationship between IgE/Syk signaling and tryptase production require significant amounts of time (>6 months). As a result, we were unable to perform new experiments to address this question within the timeframe provided to respond to the reviewers’ critiques. Nevertheless, we demonstrate that genetic deficiency in IgE (Figure 4A-D), depletion of IgE with anti-IgE antibody (Figure 4E-H), or pharmacologic inhibition of Syk (Figure 5E-I), all protect mice against development of osteoarthritis following DMM. We now address this limitation in the Discussion of our manuscript:

Limitations paragraph: “There are multiple potential limitations to this study… Additionally, demonstration in vivo that IgE/Syk signaling is associated with increased tryptase production would further strengthen our findings.”

10) The author must examine whether DMM induces increases in IgE concentration in synovial fluids in comparison to those of sham operation. If so, is it sufficient to stimulate synovial mast cells?

We agree with the reviewer that it would be interesting to compare IgE levels in DMM and sham-operated mouse joints. Prior data indicate that levels of bulk (total) IgE are not elevated in the synovial fluids of human osteoarthritic joints (Hunder and Gleich, 1974). In addition, as described in our response to the Reviewing Editor’s comment #3 (above), in preliminary studies we found that bulk IgE levels in both synovial fluids and sera are not significantly elevated in individuals with osteoarthritis. As described above, we did observe a trend towards increased levels of bulk IgE in sera derived from individuals with osteoarthritis as compared to individuals with healthy joints. Further, our preliminary data suggest that antigen-specific IgE might be present in human osteoarthritis synovial fluid and thereby might play a role in osteoarthritis pathogenesis (preliminary data provided in response to Reviewing Editor’s comment #3, above).

While we agree that further investigation of IgE levels in mice is an interesting question, obtaining synovial fluids from mice is technically difficult – particularly from sham operated joints (e.g. mouse joints with no osteoarthritis) in which there is not significant accumulation of excess synovial fluid. As a result, we were not able to perform these experiments. As outlined above, we believe investigation of the role of antigen-specific IgE in human osteoarthritis represents a more promising line of investigation (as compared to analysis of bulk IgE in mouse synovial fluids). We now address the need for further investigation of the role of IgE and antigen-specific IgE in the Discussion:

“Previous reports found that IgE levels are not elevated in the synovial fluids of osteoarthritis joints (Hunder and Gleich, 1974), and in preliminary studies we observed only trends towards an association of increased bulk IgE in serum with osteoarthritis in humans. It remains possible that antigen-specific IgE contributes to mast cell activation in osteoarthritis, and future studies will be needed to investigate the relative contributions of antigen-specific IgE and other mechanisms of mast cell activation in the pathogenesis of osteoarthritis.”

11) The authors must examine whether the DMM-induced cartilage destruction as well as the related signatures (proinflammatory cytokines, proteases expression, fibroblast proliferation and soluble glycosaminoglycans) are canceled in the Cpa3-Cre Syk-flox mice.

Given that FcR1 signals via Syk (Benhamou et al., 1993), the experiment recommended above would further confirm a role for the FcR1 /Syk pathway in the DMM model of cartilage destruction. Based on our demonstration that both genetic deficiency of FcR1 (Figure 5A-D) and pharmacologic inhibition of Syk (Figure 5E-H) prevent cartilage degradation in DMM mice, we believe that this additional experiment is redundant and would not significantly advance the current findings. Nevertheless, we now provide new data on the transcriptional expression of multiple mast cell mediators, including pro-inflammatory cytokines and proteases, in the joints of DMM mice following treatment with the Syk-inhibitor PRT062607. Six-weeks following DMM, transcriptional expression of IL-1, CCL2, CCL5, IL-6, TNF, MMP13, and ADAMTS4 was significantly reduced in the joints of PRT062607-treated as compared to vehicle-treated mice. These data are now included as new Figure 5I (see response above to Major Concern #6 from the Reviewing Editor).

[Editors' note: further revisions were suggested, as described below.]

As you can see below, one reviewer had some concerns regarding the correlation between IgE level and severeity of osteoarthritis. Regarding this concern, I judge the Ig-7KO, FceRI KO, and anti-IgE data to provide convincing genetic evidence, and as mentioned in your letter, that authors' explanation of why the IgE level did not significantly change is reasonable. Moreover, the authors carefully mentioned, for instance, "using genetic and pharmacological approaches" in the Abstract.Please address the minor concerns of Reviewer #2 outlined below.

Reviewer #1:

The authors have answered a good part of the concerns of the reviewers. The manuscript is improved and I recommend that it will be accepted for publication.

Reviewer #2:

The authors proposed that the mast cell activation mediated by IgE/FceRI/Syk signaling is responsible for pathogenesis of OA. The authors corrected some statements in the revised manuscript. However, as I mentioned below, the fundamental problems of this manuscript have not yet been solved. In addition, they did not perform additional experiments to strengthen their conclusion.

As part of our prior response to review, we performed multiple additional experiments including new in vivo experiments that provided further mechanistic data demonstrating a key role for IgE/FceRI/Syk signalling in mast cells in the pathogenesis of OA.

First, following DMM, we treated mice with either the Syk inhibitor PRT062607 or vehicle for 6-weeks, and then harvested DMM joint synovial tissue and used qPCR to analyze expression levels of mRNAs encoding pro-inflammatory cytokines and proteases. We found that expression of multiple pro-inflammatory cytokines, chemokines and proteases were reduced in DMM joint tissues derived from mice receiving pharmacologic treatment with the Syk inhibitor as compared to vehicle. These data are included in new Figure 5I.

Second, we provided new in vivo data demonstrating a key role for the high-affinity IgE receptor, Fcer1 (*Fcer1^-/-^*), on the development of osteoarthritis following medial meniscectomy (MM). Following MM, *Fcer1^-/-^* mice were protected from cartilage degeneration as compared to wild-type mice (new Figure provided in prior Response to Review). This data recapitulates our results in the DMM model (Figures 4 and 5), indicating that IgE-signaling promotes osteoarthritis pathogenesis in multiple mechanical injury models.

Further, we believe our conclusions are well supported by our data. Specifically, we believe that our Ig-7 KO, FceR1 KO, and anti-IgE data provide convincing genetic and pharmacologic evidence that IgE and IgE/FceRI/Syk signaling plays a critical role in the pathogenesis of osteoarthritis.

The authors showed that bulk IgE level in serum has no correlation with severity of osteoarthritis.

The reviewer is correct that our data suggest that there is not a strong correlation between bulk IgE in serum and the severity of OA – in a cohort of 1200 OA patients, we observed only trends towards increased bulk IgE in the serum of patients with more severe OA (based on KL score) (P = 0.08). We previously added the following text to the Discussion to address this preliminary finding as well as possible explanations:

“Previous reports found that IgE levels are not elevated in the synovial fluids of osteoarthritis joints (Hunder and Gleich, 1974), and in preliminary studies we observed only trends towards an association of increased bulk IgE in serum with osteoarthritis in humans. It remains possible that antigen-specific IgE contributes to mast cell activation in osteoarthritis, and future studies will be needed to investigate the relative contributions of antigen-specific IgE and other mechanisms of mast cell activation in the pathogenesis of osteoarthritis.”

ELISA analysis also demonstrated that synovial fluid IgE level in patients with OA was lower than that of patients with RA and PsA. (In addition, the ELISA analysis lacks appropriate controls such as healthy control.) These results made me feel that it is very unlikely that IgE is related to development of OA.

As discussed above and in our prior response to review, we and others found that levels of total IgE in either sera or synovial fluids are not associated with osteoarthritis risk or severity in humans. While increased serum levels of IgE can be detected in some atopic diseases, this is also not always the case (Novak and Bieber, 2003) – demonstrating that even in diseases well established to be mediated by IgE (e.g. antigen-specific IgE), elevations in bulk IgE are not always present. Importantly, in such allergic diseases it is antigen-specific IgE, and not bulk IgE, that plays a critical role in activating mast cells. Thus, it is possible that antigen-specific IgE, rather than bulk IgE, might play a role in the pathogenesis of osteoarthritis; or that antigen-independent mechanisms are responsible for IgE-mediated mast cell activation in osteoarthritis. Thus, we are unsure of the significance of differences in bulk IgE levels between different diseases. We do not believe these potential differences are relevant to the mechanisms by which IgE contributes to the pathogenesis of osteoarthritis.

We believe our conclusions are already well supported by our data. Specifically, we believe that our Ig-7 KO, FceR1 KO, and anti-IgE data provide convincing genetic and pharmacologic evidence that IgE plays a critical role in the pathogenesis of osteoarthritis.

The authors also referred to the possibility that antigen specificity of IgE rather than the amount of IgE may be involved in development of OA by showing the preliminary data. However, the data are too preliminary to convince us of the surprising finding that OA is an autoimmune disease caused by IgE against join antigen(s) and mast cells. The proposed mechanism is not well based on evidence. Taken together, I cannot recommend this manuscript for the publication in eLife.

As we stated in our prior response to review, we believe our initial data on antigen-specific IgE is too preliminary to include in the manuscript and as a result we have not included it – we agree with the reviewer on this point.

However, we respectfully disagree that our findings suggest that osteoarthritis is an autoimmune disease caused by IgE against joint antigens(s). A role for antigen-specific IgE, either against allergens or cartilage/synovial tissue breakdown products, is one of several potential mechanisms by which IgE could contribute to mast cell activation in synovial tissues and thereby the development of OA. We have now added to the following paragraph in the Discussion to address this consideration:

“While many classical IgE-mediated allergic diseases including asthma, allergic rhinitis and eczema exhibit comorbidities (Pedersen and Weeke, 1983), we are not aware of evidence of a clear link between osteoarthritis and allergic diseases. There is significant evidence that these classic allergic diseases are caused by antigen-specific IgE-dependent activation of mast cells (Galli and Tsai, 2012; Hamelmann et al., 1997; Oettgen and Geha, 2001; van der Heijden et al., 1993). If the role of mast cells in osteoarthritis pathogenesis is dependent on antigen-specific IgE, the target antigens could potentially be exogenous allergens. Another possibility is that the IgE target antigens are bone or cartilage breakdown products that give rise to neoantigens following joint injury or instability. Although we are not aware of evidence that osteoarthritis is associated with classic allergic and/or IgE-dependent diseases, an epidemiological analysis that addresses this important question is warranted. Antigen-specific antibody responses can also be generated in autoimmune responses, however OA synovial linings do not exhibit histologic features consistent with an adaptive autoimmune response and we do not believe our findings suggest the presence of an classical adaptive autoimmune response. Future studies are needed to determine if IgE targets specific antigens, and to further characterize the role of IgE, in osteoarthritis.”

1) In general, OA is induced by long lasting mechanical stresses and can be accompanied by chronic inflammation, whereas IgE mediates antigen-specific acute and drastic inflammatory responses in allergic diseases. Thus, it is not easy to imagine that IgE-mediated immune responses are implicated in OA pathogenesis and induction of chronic inflammation that accompanies OA. The authors have to give convincing explanation.

We agree that long-lasting mechanical stresses likely contribute to the pathogenesis of osteoarthritis in a significant subset of patients, and added the following text to the Discussion to address this point: “Mechanical instability and stresses likely contribute to the pathogenesis of osteoarthritis in a significant subset of patients. We do not believe that a role for mechanical stresses is inconsistent with a role for IgE and mast cells, and it is possible that mechanical stresses produce cartilage breakdown products and/or cellular responses that promote IgE-dependent mast cell activation.”

2) Authors showed that IgE level does not corelate with OA onset or severity. I think that this is contradictory to authors' conclusion. Authors tried to explain this by the antigen-specific IgE, but it is unbelievable that OA is an autoimmune disease caused by IgE against join antigen(s). If so, they should identify the specific antigens recognized by IgE from the synovial tissues.

As described above, there are examples of IgE-mediated allergic diseases in which bulk IgE levels do not correlate with disease severity, and this may also be the case in osteoarthritis. As described in our responses above, we do not believe our findings suggest that osteoarthritis is an autoimmune disease and added several sentences to the Discussion to address this possibility (see above for full response and new text added). We believe definitive characterization of the potential targets of IgE in osteoarthritis, and/or if other mechanisms promote IgE-dependent mast cell activation in osteoarthritis, is beyond the scope of the current manuscript.

3) The authors could not analyze synovial derived mast cells by flow cytometry because enzymatic dissociation of synovium physically destroys mast cells. But many researchers succeeded in cell preparation from human and mouse synovial tissue. In particular, FACS analysis of mast cells in synovial tissues was previously shown (Lee et al., Arthritis Rheum. 2013). Therefore, the authors must show the accurate number of mast cell by FACS in Figure 1 to 5 and Figure 2—figure supplement 1.

We demonstrate that mast cells are present in human OA synovial lining by (i) immunostaining (New Figure 1—figure supplement 2), and (ii) electron microscopy (Figure 1B and C). In New Figure 1—figure supplement 2 we quantitate mast cells in human OA synovial linings and demonstrate that they represent approximately 1% of cells in the synovial lining.

The work of Lee et al. is already cited in the following sentence in the Introduction of our manuscript: “Several studies have documented the presence of mast cells and their mediators in the synovium and synovial fluids of individuals with osteoarthritis (Buckley et al., 1997; Dean et al., 1993; Lee et al., 2013).” We appreciate that the reviewer comments on the findings of Lee et al. (Arthritis Rheumatology, 2013), as their findings as well as the findings of the other groups cited all demonstrate the presence of mast cells in human osteoarthritis synovial linings and thereby support the findings presented in our manuscript. Further, these prior studies used multiple different experimental approaches and methods to demonstrate the presence of mast cells in human OA synovial linings – thus, these prior observations are consistent with and further support the findings reported in our manuscript. Even if technically feasible using other tissue digestion methods, we do not believe additional flow cytometry studies to analyze synovial mast cells would meaningfully strengthen or add to the findings presented in our manuscript.

4) The authors cannot conclude that mast cell-derived tryptases promote osteoarthritis-associated pathology, since it remained unclear whether tryptase secreted by mast cells is sufficient to induce the destruction of the synovial cartilage. As I pointed out in the comment #6, addition of huge amount of tryptase is too artificial. The authors have to perform the co-culture experiment of mast cells with human cartilage explants. Furthermore, the authors have to show the evidence that chondrocyte undergoes apoptosis as I mentioned in comment #8.

We appreciate the reviewer’s questions, and now add the following text to the Discussion to address these considerations. “Although we have not formally demonstrated that mast cells are the source of tryptases that promote the pathogenesis of OA, our data demonstrate that: (i) mast cells comprise 1 – 3% of synovial cells in both OA and PT non-OA synovial linings, (ii) mast cells are actively degranulating to release tryptase in human OA synovial linings (Figure 1B and C), and pharmacologic inhibition of typtase prevents the development of OA in mice (Figure 3A-D). Mast cells are the predominant producer of tryptases, and to a lesser extent basophils (Schwartz, 2006), and histologic examination of human and mouse OA synovial linings and tissues for basophils did not reveal basophils. Together, these data suggest that mast cells are the cellular source of tryptase in human and murine OA.”

Regarding this reviewer’s prior comment #6, we agree with the reviewer that the in vitro human cartilage explant experiments represent a highly artificial experimental system. As a result, in the prior revision we removed the data and figure panels in which relatively large amounts of tryptase were incubated with cartilage explants (original Figures 3A and B – now removed). We note that in vivo treatment of DMM mice with the tryptase inhibitor, APC366, preserves cartilage (Figure 3A and B) – demonstrating that tryptases do promote cartilage degradation in osteoarthritis. In addition, as part of the prior revision, we performed new in vivo experiments that demonstrated that pharmacologic inhibition of tryptases results in decreased transcriptional expression of inflammatory and degradative mast cell mediators in DMM joint tissues (new Figure 3E). Specifically, we demonstrated that 6 weeks following DMM that treatment of mice with the tryptase inhibitor APC366 reduced the transcriptional expression of multiple inflammatory and degradative mediators including IL-1, IL-6, IL-8, CCL2, CCL5, ADAMTS4 and *MMP3*.